# Non-vesicular phosphatidylinositol transfer plays critical roles in defining organelle lipid composition

Yeun Ju Kim [1], Joshua G Pemberton [1], Andrea Eisenreichova[2], Amrita Mandal[1], Alena Koukalova[1], Pooja Rohilla[1], Mira Sohn [1], Andrei W Konradi[3], Tracy T Tang [3], Evzen Boura[2] & Tamas Balla [1]✉

## Abstract

**Phosphatidylinositol (PI) is the precursor lipid for the minor phosphoinositides (PPIns), which are critical for multiple functions in all eukaryotic cells. It is poorly understood how phosphatidylinositol, which is synthesized in the ER, reaches those membranes where PPIns are formed. Here, we used VT01454, a recently identified inhibitor of class I PI transfer proteins (PITPs), to unravel their roles in lipid metabolism, and solved the structure of inhibitor-bound PITPNA to gain insight into the mode of inhibition. We found that class I PITPs not only distribute PI for PPIns production in various organelles such as the plasma membrane (PM) and late endosomes/lysosomes, but that their inhibition also significantly reduced the levels of phosphatidylserine, di- and triacylglycerols, and other lipids, and caused prominent increases in phosphatidic acid. While VT01454 did not inhibit Golgi PI4P formation nor reduce resting PM PI(4,5)P$_2$ levels, the recovery of the PM pool of PI(4,5)P$_2$ after receptor-mediated hydrolysis required both class I and class II PITPs. Overall, these studies show that class I PITPs differentially regulate phosphoinositide pools and affect the overall cellular lipid landscape.**

**Key words** Phosphatidylinositol; Non-Vesicular Lipid Transport; Membrane Contact Sites; Phospholipase C; Golgi Complex
**Subject Categories** Membranes & Trafficking; Organelles

## Introduction

Eukaryotic cells organize their metabolic processes in membrane-enclosed organelles that all feature unique lipid compositions. While most lipids are synthesized in the endoplasmic reticulum (ER), there is a steady-state enrichment of specific lipids in other organelle membranes, and, therefore, lipids must be selectively transported out from the ER to other organelles. While lipid transport through vesicular membrane trafficking can deliver bulk lipids between organelles, non-vesicular lipid transport has proven to be an important means by which cells can rapidly distribute membrane lipids (Lipp et al, 2020; Prinz et al, 2020). In particular, while

phosphoinositides (PPIns), which are the phosphorylated derivatives of phosphatidylinositol (PI), represent a quantitatively small fraction of total membrane phospholipids, they play critical roles in controlling many aspects of membrane dynamics and cellular signaling (Balla, 2013). Most recently, PI 4-phosphate (PI4P) gradients, which are formed between adjacent membrane compartments, have been identified as important drivers for the non-vesicular transport of structural lipids, including cholesterol (Mesmin and Antonny, 2016), phosphatidylserine (PS) (Chung et al, 2015; Maeda et al, 2013; Moser von Filseck et al, 2015), and perhaps other lipid classes. Whereas PI4P, which is produced by four different PI 4-kinase (PI4K) enzymes that are selectively localized to specific organelles (Boura & Nencka, 2015; Waugh, 2019) was initially believed only to serve as the precursor of the important plasma membrane (PM)-enriched PPIn lipid, PI 4,5-bisphosphate [PI(4,5)P$_2$], it is now understood that PI4P also regulates vesicular trafficking at the Golgi complex and in endosomes (Baba and Balla, 2020; D'Angelo et al, 2012; Waugh, 2019). Consequently, all these transport processes must depend on the production of PI in the ER and its subsequent delivery to the organelle membranes, where the PI4K enzymes locally convert it to PI4P.

It has been widely assumed that PI-transfer proteins (PITPs) deliver PI from the ER to the various other organelles. PITPs were first identified in the late 1960s, when it was shown that soluble tissue extracts contain proteins that can exchange phospholipids between membranes of the mitochondria and microsomes in vitro (Wirtz and Zilversmit, 1969). The preferred lipids transferred by these proteins were PI and, to a lesser degree, phosphatidylcholine (PC), hence the initial name of PI/PC transfer proteins, or PITPs. PITPs were soon purified from the bovine brain (Helmkamp et al, 1974) and subsequently cloned (Dickeson et al, 1989). PITPs were functionally identified as soluble protein factors that were necessary to restore GTPγS-stimulated phospholipase C (PLC)-mediated production of inositol phosphates in permeabilized HL60 cells (Thomas et al, 1993). Parallel studies also found PITP as one of three soluble factors required for priming secretory vesicles for exocytosis in PC12 cells (Hay and Martin, 1993). However, despite the extensive work done on PITP proteins, including their functional homolog, Sec14 in yeast (Ashlin et al, 2021; Cockcroft and Garner, 2011; Grabon et al, 2019; Grabon et al, 2015; Lev, 2010), direct proof that PITPs are indeed responsible for delivering PI to various organellar membranes has been lacking, partially due to functional redundancies between the various PITPs that are found in higher organisms.

[1]Section on Molecular Signal Transduction, Eunice Kennedy Shriver, National Institute of Child Health and Human Development, National Institutes of Health, Bethesda, MD 20892, USA. [2]Institute of Organic Chemistry and Biochemistry of the Czech Academy of Sciences, Flemingovo nam. 2., 166 10 Prague 6, Czech Republic. [3]Vivace Therapeutics, San Mateo, CA 94404, USA. ✉E-mail: ballat@mail.nih.gov

Mammalian PITPs belong to one of two classes: class I PITPs are small proteins (~30 kDa) encoded by two separate genes, PITPNA and PITPNB, with the latter producing two splice forms that differ at their C-termini (Ashlin et al, 2021). Class II PITPs, such as Nir2 and Nir3 (or PITPNM1 and PITPNM2), are larger proteins that also have a canonical PITP domain at their N-termini, which is analogous to the class I PITP-fold and followed by several additional domains that mediate interactions with membranes as well as other proteins (Raghu et al, 2021). Importantly, genetic silencing of the individual PITPs in model organisms yielded phenotypes that could not be unequivocally traced to specific defects in PI-transfer (Alb et al, 2003; Alb et al, 2002; Alb et al, 2007; Huang et al, 2018; Xie et al, 2018; Zhao et al, 2023; Zhao et al, 2017). In a similar fashion, cellular studies with genetic targeting of individual PITPs yielded very subtle phenotypes and little or no change in PPIns levels (Alb et al, 2002; Carvou et al, 2010; Kim et al, 2022). Ideally, PITPs would need to be inactivated in a rapid fashion to allow for assessment of their direct contributions to the regulation of various subcellular lipid pools. The recently identified natural compound microcolin B and its more active derivative, VT01454, which selectively inhibits the class I PITPs (Li et al, 2022), now permits these types of studies for the first time.

In the present study, we report on the effects of acute, pharmacological inhibition of class I PITPs on the various PPIn-rich membrane compartments as well as on the overall cellular lipidome using intact cells. We specifically show that acute pharmacological inhibition of class I PITPs not only leads to profound changes to the PI4P levels in the PM and late endosomes/lysosomes, but also results in the accumulation of selected molecular species of phosphatidic acid (PA) as well as causes a reduction of cellular diacylglycerol (DAG), triacylglycerol (TAG) and PS. Contrary to expectations, inhibition of class I PITPs failed to alter PI4P levels in the Golgi complex, or impact the resting levels of $PI(4,5)P_2$ in the PM, while also only having small effects on PI3P in early endosomes. Moreover, our pharmacological studies show that the recovery of $PI(4,5)P_2$ within the PM after a strong PLC activation required both the class I and class II PITPs. Finally, we solved the structure of the human PITPNA bound to the inhibitory compound, VT01454, and performed structural analysis to better understand the molecular mechanism that governs the interaction of the class I PITP proteins with cellular membranes. These results show that PITPs do play important roles in the specific maintenance of the PI4P pools in the PM and late endosomes, but our pharmacological approach also highlights the tight coupling between non-vesicular PI-transport processes and the integrated activity of the lipid synthetic and distribution machineries.

## Results

### Pharmacological inhibition of class I PITPs selectively affects distinct subcellular PPIns pools

Individual knockout (K/O) of either PITPNA or PITPNB in HEK293 cells (Kim et al, 2022) failed to appreciably change the size of the PI4P or $PI(4,5)P_2$ pools, as assessed by prolonged labeling with myo-[$^3$H]inositol (Fig. 1A). Selective PITP knockout also did not alter the recovery rate of these PPIn lipids after PLC activation by angiotensin II (AngII) stimulation (Kim et al, 2022). With the availability of the PITP inhibitor, VT01454, we wanted to test how

acute inhibition of both class I PITPs affects the various subcellular pools of PPIns. Treatment of HEK293 cells prelabeled with myo-[$^3$H]inositol for 24 h with VT01454 for 30 min substantially reduced the signal for total PI4P but not in $PI(4,5)P_2$ (Fig. 1A). To understand the effects of the inhibitor on the PI lipids of various organelles, we used genetically encoded lipid binding probes (Hammond et al, 2022) and analyzed the localization of these sensors both by confocal microscopy and bioluminescence resonance energy transfer (BRET)-based measurements (Fig. 1B) (Toth et al, 2016) in HEK293-AT1 cells that stably express the rat AT1 AngII receptors (Hunyady et al, 2002). Importantly, BRET analysis allows for the assessment of organelle-specific changes in selected lipids at the cell population level. Briefly, a protein module with specific lipid recognition (or lipid binding domain, LBD) is fused with a Super Renilla (or other forms of) luciferase (sLuc) and transfected together with an mVenus protein targeted to the organelle of choice (usually from a single vector). The presence of the lipid in the targeted membrane attracts the LBD, which brings sLuc in close proximity to the organelle-anchored mVenus and, in the presence of a suitable sLuc substrate, allows for energy transfer between these labels to yield excitation of mVenus. The extent of energy transfer depends on the level of the particular lipid in the organelle membrane in question (Fig. 1B and see (Toth et al, 2019) for more details on BRET).

First, we tested the effects of VT01454 on the PM levels of PI. For this, we used the bacterial $Bc$PI-PLC$^{H82A}$ probe described earlier in the BRET analysis (Pemberton et al, 2020). In that study, we showed that the low level of PI in the PM of resting cells can be acutely increased when the PM-resident PI4KA enzyme is inhibited using the selective inhibitor, GSK-A1 (Bojjireddy et al, 2014). However, after VT01454 (100 nM) pretreatment for 30 min, the rise in PM level of PI after blocking PI4KA was almost completely abolished. (Fig. 1C). Next, we analyzed the PI4P pools within the PM using the PI4P sensor, (2x)P4M (Hammond et al, 2014) using BRET. This analysis showed that the PM pool of PI4P was rapidly reduced after the addition of VT01454 with a near-maximal effect at a concentration of 100 nM. VT-treatment was almost as effective in reducing PM levels of PI4P as the inhibition of PI4KA (Fig. 1D). The basal level of $PI(4,5)P_2$ was only minimally affected by treatment with VT01454 in spite of the large decrease observed in the PM level of PI4P (Fig. 1E). These BRET results were consistent with those using myo-[$^3$H]inositol labeling.

Next, we investigated the effects of VT01454 on the intracellular pools of PPIns. Earlier studies have shown that PI4P is present in Rab7-positive endosomes (Baba et al, 2019; Hammond et al, 2014), while PI3P is enriched in Rab5-positive early endosomes (Simonsen et al, 1998). When the effect of VT01454 on the PI4P pool associated with the Rab7-positive compartment was studied, the true effect of the inhibitor was initially masked by the release of the PI4P sensor from the PM and its redistribution to endomembrane compartments, which resulted in an increased BRET signal [as shown in (Baba et al, 2019) when the PI4KA was inhibited]. However, this signal did not remain elevated, but instead showed a sharp decline below the resting BRET levels (Fig. 1F, left panel). This decline suggested that the Rab7 compartment is losing most of its PI4P in the presence of the PITP inhibitor. This was further examined using a different approach. We have previously reported that inhibition of OSBP-mediated PI4P/cholesterol exchange by the OSBP inhibitor, OSW1, caused a rapid increase

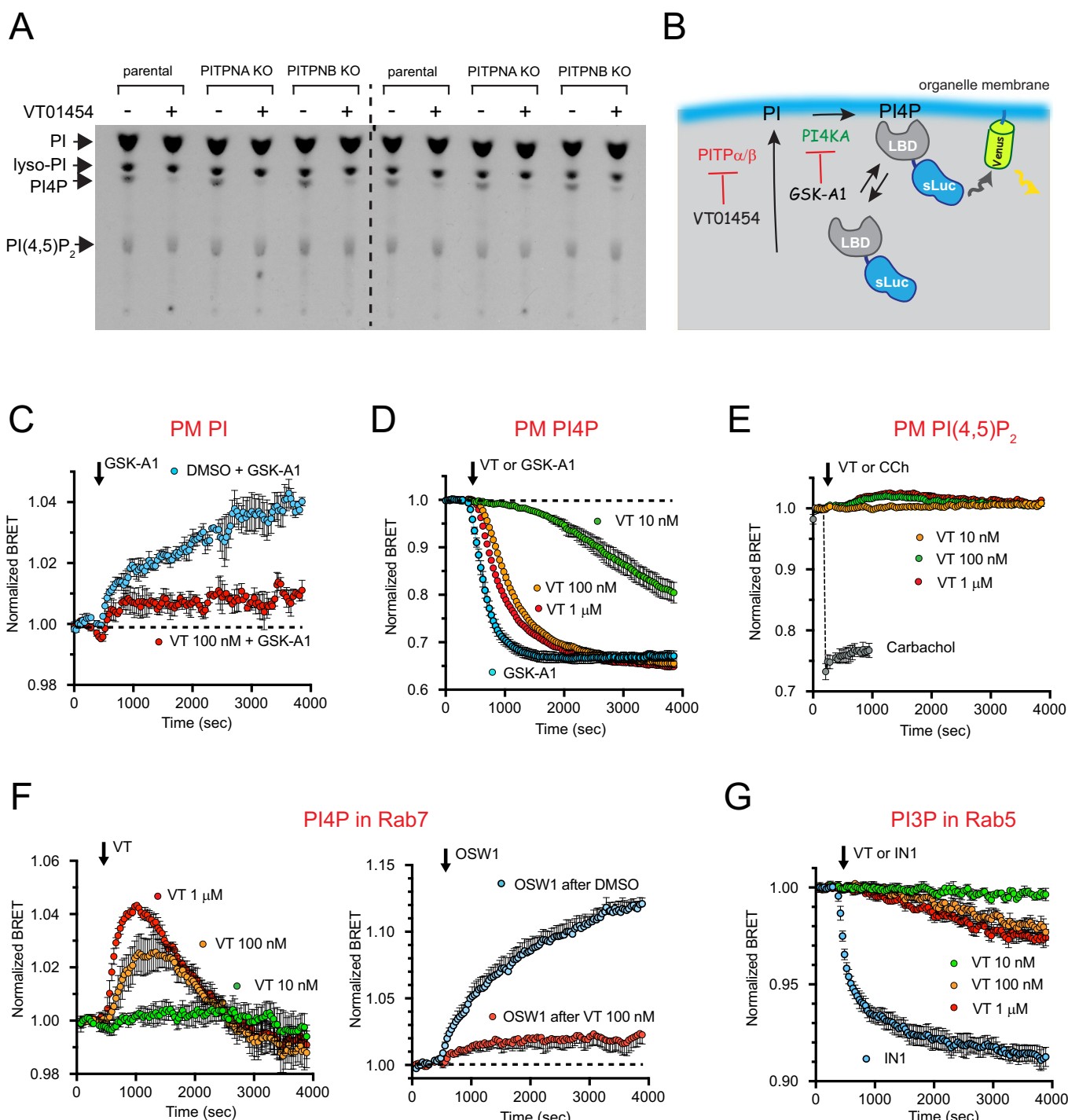

in the level of PI4P in the Rab7-positive compartment (Baba et al, 2019). However, when the cells were pretreated with VT01454 (100 nM) for 30 min and then treated with OSW1 (30 nM), the increase was greatly inhibited (Fig. 1F, right panel). In contrast, only a minor and slow effect of VT01454 treatment was observed on the PI3P pool that is present in Rab5-positive compartments, which was apparent when the effect of VT01454 was compared to the change caused by inhibition of the resident PI 3-kinase, Vps34 (Fig. 1G).

These data collectively suggested that PI delivery by class I PITPs is critical for the maintenance of the PI4P levels in the PM and Rab7-positive endosomes, two compartments where steady-state PI levels are found to be low. In contrast, PI3P levels in the Rab5-positive endosomes are more resistant to PITP inhibition, and the resting $PI(4,5)P_2$ levels in the PM can be maintained even at very low levels of PM PI4P, as already reported earlier when PI4KA was acutely inhibited (Bojjireddy et al, 2014; Gulyas et al, 2022; Hammond et al, 2012).

◄ **Figure 1. Different cellular phosphoinositide pools are affected by inhibition of PITPs by VT01454.**

(A) TLC analysis of lipid extracts from HEK293-AT1 cells labeled with *myo*-[³H]inositol for 24 h and treated with VT01545 (100 nM) for the last 30 min of the labeling period. Note that knockout (K/O) of individual PITPs show no obvious effect on the labeling of any of the PI lipids, and that VT01454 strongly inhibited the labeling of PI4P but not PI or PI(4,5)P₂ in each of the cell lines. (B) Cartoon depicting the principle of lipid detection in specific membranes using the BRET principle. Cells are transfected with a luciferase enzyme (sLuc) conjugated to a lipid binding domain (LBD) specific to the lipid in question, together with the Venus protein targeted to a specific organelle membrane. Resonance energy transfer occurs between the sLuc and Venus in the presence of a luciferase substrate, only when the lipid in the membrane attracts the LBD-sLuc conjugate. This method can monitor lipid changes in specific organelle membranes using cell populations. (C) Accumulation of PI in the PM after the addition of the PI4KA inhibitor, GSK-A1 (100 nM) (blue), and its inhibition by 30 min VT01454 (100 nM) treatment (red). BRET analysis using the *bc*PI-PLC^H82A (Pemberton et al, 2020) as the LBD and values normalized in each case to the respective DMSO-treated controls. Grand average ± SEM from three independent experiments performed in triplicates. Source data are available online for this figure. (D) Dose-dependent inhibition of PI4P in the PM by VT01454 or GSK-A1 (100 nM). BRET analysis using the (2x) P4M as the LBD to monitor PI4P. BRET values are normalized to the DMSO-treated control. Grand average ± SEM from three independent experiments, each performed in triplicates. Source data are available online for this figure. (E) Lack of inhibitory effect of VT01454 on PM PI(4,5)P₂ levels. BRET analysis using the PLCδ1-PH domain as the PI(4,5)P₂ LBD. BRET values were normalized to the DMSO-treated control. For comparison, we plotted the response to stimulation of cells expressing the M1 muscarinic receptor with carbachol (gray). Grand average ± SEM from three independent experiments, each performed in triplicates. Source data are available online for this figure. (F, Left) Dose-dependent effect of VT01454 on PI4P monitored in the Rab7 compartment. BRET analysis using the (2x)P4M PI4P sensor paired with Rab7-targeted mVenus. BRET values were normalized to the DMSO-treated control. Grand average ± SEM from three independent experiments performed in triplicates. Source data are available online for this figure. Note the initial increase in the BRET signal that is due to the liberation of the PI4P reporter from the PM and its association with the Rab7 compartment. However, this rise is transient, followed by a rapid decrease indicating the loss of PI4P due to the lack of PI delivery to the Rab7 compartment. (F, Right), PI4P in the Rab7 compartment is increased after blocking the PI4P/cholesterol exchanger, OSBP, with OSW1 (30 nM). This increase is greatly reduced in cells pretreated with VT01454 (100 nM) for 30 min. BRET analysis was as described for F, left panel. Grand average ± SEM from three independent experiments, each performed in triplicates. Source data are available online for this figure. (G) Dose-dependent inhibition of PI3P generation in the Rab5-positive compartment by VT01454 or by the Vps34 inhibitor, IN-1 (300 nM). BRET analysis using the (2x)FYVE^Hrs as the PI3P LBD, paired with the Rab5-targeted Venus. BRET values normalized to the DMSO-treated control. Grand average ± SEM from three (VT01454) or four (IN-1) independent experiments, each performed in triplicates. Source data are available online for this figure.

## Inhibition of class I PITPs does not acutely reduce P4P levels in the Golgi complex

It has been widely assumed that PITPs deliver PI to the Golgi complex for the maintenance of PI4P levels (Grabon et al, 2019). However, unexpectedly, the Golgi-specific pool of PI4P was not affected by the VT01454 compound (100 nM) during a 30-min treatment regardless of which Golgi-restricted PI4P sensor (FAPP1-PH, FAPP2-PH, CERT-PH, or GOLPH3) was used (Fig. 2A–K). BRET analysis of the Golgi compartment has been notoriously difficult, most likely because of the dynamic cycling of the Golgi targeting sequences between the Golgi complex and the other membrane compartments, including the ER, endosomes as well as PM. Therefore, for these experiments, the quantification of PI4P levels was based on kinetic recordings captured by confocal microscopy. Some of the PI4P binding domains, including FAPP2-PH, CERT-PH, or GOLPH3, actually showed a slightly enhanced Golgi localization following the VT01454 treatment (Fig. 2G,I,K). To determine whether the localization of these PI4P binding proteins requires constant conversion of PI to PI4P, we used an inhibitor of PI4KB, MI-14 (5 μM) (Mejdrova et al, 2015), and tested its effect on FAPP1-PH localization. As reported previously, the levels of PI4P at the Golgi showed oscillations after PI4KB inhibition, which is thought to be due to the contribution of PI4P generated by PI4K2A (Mesmin et al, 2017). Indeed, MI-14 (5 μM) caused a decrease in FAPP1-PH localization to the Golgi, with many cells showing an oscillatory pattern (Fig. 2D). In PI4K2A knockout cells (Baba et al, 2019), MI-14-treatment caused a larger overall decrease in FAPP1-PH localization, but some cells still showed oscillatory changes (Fig. 2E). To determine whether VT01454 treatment limits the ability of the Golgi complex to further increase PI4P levels even if the steady-state level of PI4P is not reduced by the inhibitor, we subjected VT01454-treated cells to treatment with OSW1. As described above, OSW1 inhibits the OSBP-mediated transport of PI4P from the Golgi complex to the ER and, hence, increases the Golgi level of PI4P in a manner that

still requires PI phosphorylation (Mesmin et al, 2017). Treatment with OSW1 (30 nM) was still able to further increase the levels of Golgi PI4P as assessed by the FAPP1-PH domain, even in cells pretreated with VT01454 [100 nM, (Fig. 2C)].

These data collectively suggested that while Golgi-associated PI4P depends on the continued conversion of PI to PI4P by Golgi-localized PI4Ks, inhibition of class I PITPs does not acutely decrease Golgi PI4P levels during the period examined (up to 40 min). Since it has been previously suggested that PITPs are required for PI4K enzymes to effectively utilize their PI substrate (Schaaf et al, 2008), another important corollary of our findings is that PI4K enzymes in the Golgi still can efficiently convert PI to PI4P when the class I PITPs are inhibited.

## Expression of drug-resistant PITP mutants can rescue the lipid changes caused by VT01454

Since VT01454 has been proposed to covalently react with Cys94 in class I PITPs (Li et al, 2022), we examined whether a mutation of the C94 position (C94S) would yield drug-resistant PITP variants. It is important to note that residue C94 has previously been implicated in supporting PC binding, and both the bcC94A and C94T mutants were shown to be defective in PC transfer in vitro (Carvou et al, 2010) [but not in (Tremblay et al, 2001)]. To minimize the steric alteration, we mutated C94 and examined the ability of the various C94 mutant PITPNAs to bind PI and PC. For this, we used recombinant PITPNAs mutated either in the 94 position (C94S and C94T) or in the 58 position (T58E), the latter substitution that is well known to eliminate PI-binding, and determined their ability to extract lipids from crude membranes prepared from cells prelabeled with [¹⁴C]-acetate. These experiments showed that all mutants of PITPNA, including the PI binding T58E mutant, showed impaired PC binding, with the C94T substitution having the strongest effect, while also confirming the inability of the T58E mutant to bind PI. Notably the C94S or C94T mutants showed stronger PI binding compared to the wild-type protein (Fig. EV1A,B).

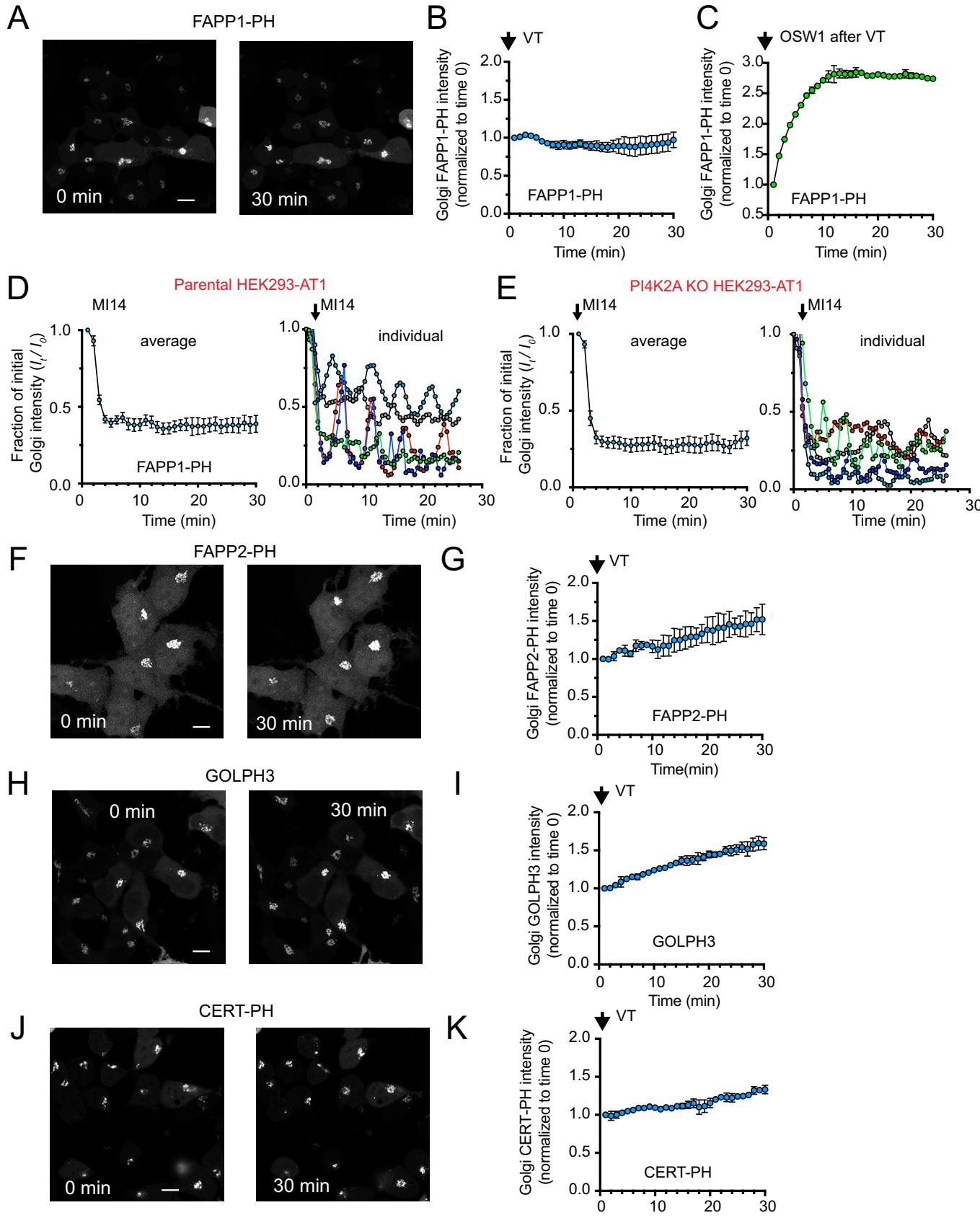

◀

**Figure 2. PI4P levels at the Golgi complex are not reduced by VT01454.**

Association of several PI4P-recognizing protein modules such as FAPP1-PH (**A–E**), FAPP2-PH (**F, G**), GOLPH3 (**H, I**), and CERT-PH (**J, K**) with the Golgi compartment was followed after VT01454 (100 nM) treatment in live cells using confocal microscopy. For details of image quantification, see Methods. Note that none of these proteins showed decreased localization after treatment, and in some cases, an increase rather than a decrease was observed. (**C**) OSW1 (30 nM)-induced increases in Golgi-associated PI4P as a result of inhibition of PI4P transport from Golgi to ER, was still obvious even after VT01454 treatment. For comparison, the change in Golgi localization of the FAPP1-PH was tested using an inhibitor of PI4KB (MI-14, 5 µM) both in parental (**D**) and in PI4K2A K/O HEK293-AT1 cells (**E**). Note the rapid drop in FAPP1-PH signal in the Golgi, that was partial and showed oscillatory behavior in the parental cells (shown in D, right panels with a selected individual traces). The inhibitor caused a more complete depletion of PI4P in the PI4K2A K/O cells, but some cells still showed oscillatory changes (**E**). Data Information: Grand averages ± S.E.M or range from four (**B**), seven (**D, E**), three (**G, I**), or two (**C, K**) independent dishes (5–15 cells, each) are shown. Representative traces of individual cells are shown on the right panels in panels (**D**) and (**E**) from one of these experiments. Scale bars 10 µm. Source data are available online for this figure.

We then tested the effects of the drug-resistant (C94S) forms of PITPs on the maintenance of the PM pool of PI4P, as this readout showed the most prominent changes in response to acute VT01454 treatment (Fig. 3A). Overexpression of the wild-type PITPNA or PITPNB altered the kinetics of the inhibitory effects associated with VT01454 treatment (100 nM; Fig. 3B), which was attributed to a reduction to the effective concentration of VT01454 by the over-expressed proteins. Alternatively, adding back PITPNA-C94S (Fig. 3C) or PITPNB-C94S (Fig. 3D) were both able to rescue the VT01454-induced reduction of PI4P levels in the PM. There was no major difference between the C94A, C94S, and C94T mutant in their ability to reverse the effects of VT01454. In contrast, the PI-binding mutant forms of the PITPNB-C94S, including either T58A or T58E mutants, showed minimal or no rescue, respectively (Fig. 3D). This agreed with earlier in vitro studies that reported a small residual PI-transfer activity for the T58A and a complete loss of PI-transfer activity for the T58E mutant (Tilley et al, 2004). We also examined whether the defects in PI delivery to the Rab7-positive compartment could similarly be rescued with the expression of the VT01454-resistant C94S mutant PITP proteins. Expression of either the inhibitor-resistant PITPNA-C94S or PITPNB-C94S restored the ability of Rab7-positive endosomes to accumulate PI4P when exposed to OSW1 (30 nM) after pretreatment of VT01454 (100 nM; Fig. 3E,F). Once again, the PI-binding mutant, T58E, was unable to rescue PITP functions in this assay. Since all C94A, C94S, and C94T mutant PITPNAs were effective in rescuing both the PM and Rab7-associated PI4P levels (Fig. 3C,D,F), it was concluded that PC transfer is unlikely to be critical for delivering PI for the PI4KA enzymes that are active in the PM and the Rab7-positive endosomes. It must be noted, though, that none of the C94 mutations completely eliminated the PC binding of the PITPNA variants tested (Fig. EV1A,B).

## Structural basis for the inhibition of PITPNA-mediated lipid transfer by VT01545

To gain a detailed structural insight into the inhibition of PITPNA by VT01454, we solved the crystal structure of the human PITPNA bound to VT01454. Briefly, for these studies, we prepared purified recombinant PITPNA and incubated it with VT01454 for 1 h before starting the crystallization trials to allow this covalent inhibitor to react with the Cys94 of PITPNA via the Thia–Michael reaction (Berne et al, 2022). We obtained crystals that diffracted to 2.3 Å resolution and solved the structure using molecular replacement with the structure of the PC-bound rat PITPNA (PDB ID: 1T27) as the search model before refining it to good geometry and R-factors (Table 1) (see more details in the Method section).

We could trace the entire polypeptide chain except for the first methionine and the last 17 amino acid residues at the C-terminus. The overall structure of the human PITPNA protein conformed with the previously described mouse, rat, and human PITP structures (Schouten et al, 2002; Tilley et al, 2004; Yoder et al, 2001); with the overall fold resembling a splayed β-sheet that is formed by eight β-strands and covered by α-helices. The lipid binding site is created by the cleft between the β-sheet and the α-helices (Fig. 4). The inhibitor is well resolved from its electron density (Fig. 4B,C). In addition to binding covalently with the reactive Cys94 residue, VT01454 also forms hydrogen bonds or water bridges with other residues that contribute to its binding. These include Gln22, which forms a hydrogen bond directly with VT01454, while residues such as Tyr18, Glu85, Thr96, His115, Lys194, and Glu217 all bind VT01454 through a water bridge (Fig. 4B).

In the previous study describing VT01454, the inhibitor was docked in the lipid binding pocket in silico using existing structures as a template, which correctly identified the interaction with Cys94 (Li et al, 2022). This approach, however, could not predict the overall structure of the PITP-fold when bound with the inhibitor as it modeled the inhibitor bound to the closed conformation of the molecule. The structure presented here, however, clearly shows its almost perfect superpositions of the VT01454-bound structure with the lipid cargo-free (apo) PITPNA structure (Schouten et al, 2002). This suggests that VT01454 binding can occur while the lipid is leaving the PITPNA in an open conformation. In this configuration, the lipid exchange α2 helix swings out, away from the lipid binding pocket (Fig. 4D). Notably, this inhibitor-bound conformation is virtually identical to the published unliganded (Apo) conformation (RMSD = 0.332 Å). Nevertheless, superpositions of the inhibitor-bound structure with those of the PI- and PC-bound PITPNA also confirmed that VT01454, indeed, occupies the space where the PI or PC molecule would be located, which explains the strong inhibitory effect associated with VT01454 treatment (Fig. 4E). Comparison of the inhibitor-bound structure with those of the PI- and PC -bound PITPs showed significantly higher RMSDs (0.616 Å and 0.546 Å for PI and PC loaded structures, respectively) and demonstrated that the positions of helices α1 and α7, similarly to helix α2, adopt the open conformation in the inhibitor-bound state (Fig. 4E).

These findings were important since the only available structure of the Apo-PITPNA showed the lipid exchange α2 helix in a position that was proposed to directly mediate membrane association, which actually resulted from the α2 helix engaging in hydrophobic interactions with another PITPNA molecule, as part of an intimate dimer (Schouten et al, 2002). The perfect superimposition of the VT01454-bound structure with this apo-PITPNA structure suggests that the α2 helix, indeed, assumes a position that is consistent with its penetrating in the membrane. Earlier studies using molecular

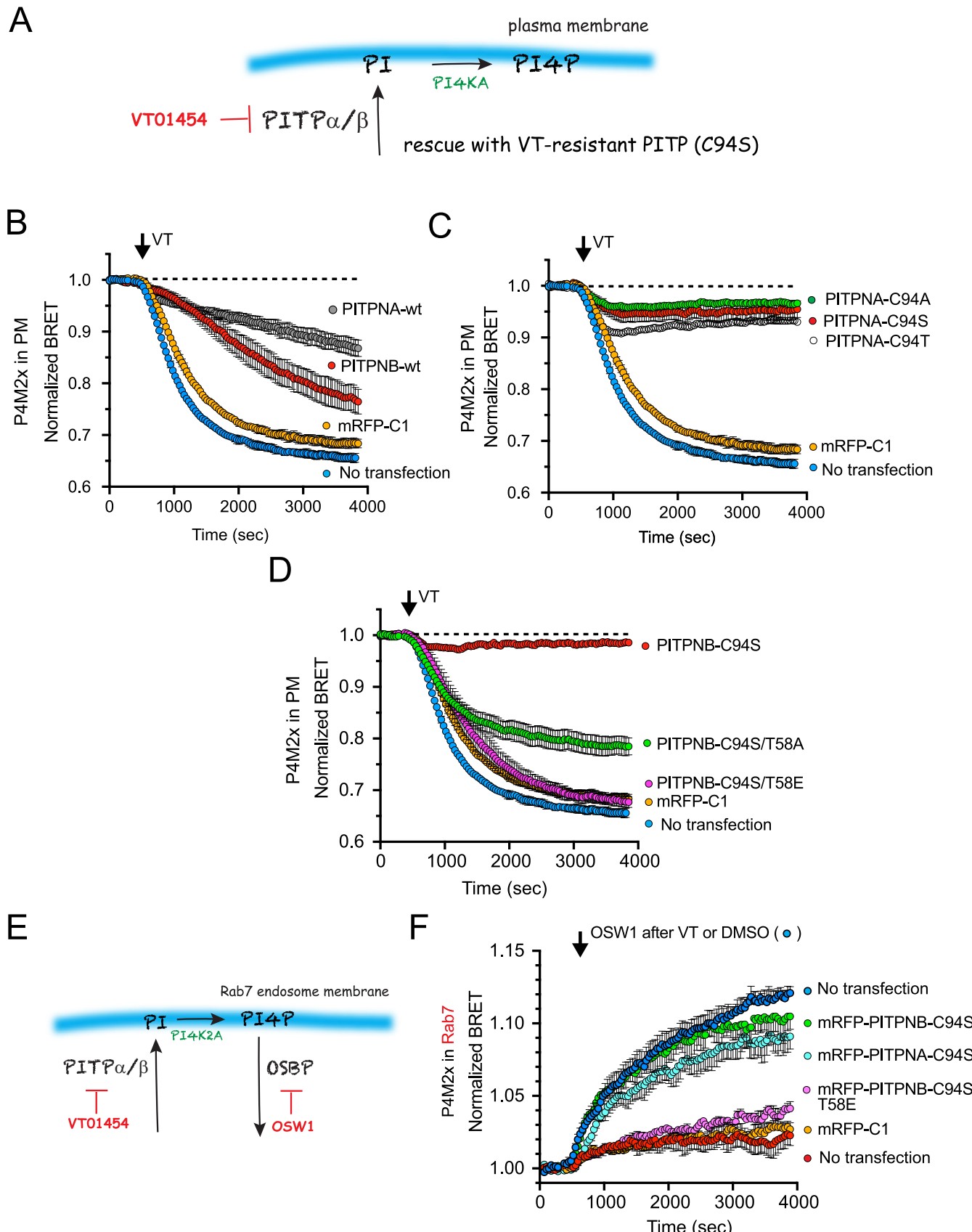

◄ **Figure 3. Expression of VT01454-resistant PITP variants overcomes PI4P depletion induced by VT01454 treatment in the PM and Rab7-positive compartments.**

(A) A cartoon showing the experimental setup for (B–D). BRET analysis was used to monitor PM PI4P using the (2x)P4M PI4P sensor paired with the PM-targeted Venus. BRET values normalized to the DMSO-treated control cells (dotted lines). mRFP-tagged PITP wild type or the indicated mutants were co-expressed with the BRET sensor, and the mRFP-C1 empty vector was used as control. The effect of the inhibitor on non-transfected cells (blue traces) is also plotted for reference in all panels. (B) Even wild-type versions of PITPNA and PITPNB proteins showed a partial rescue, as observed as a delay to the inhibition, simply because of buffering the available inhibitor. (C) PITPNA mutated in the C94 residue to either Ala, Ser, or Thr, all completely counter the inhibition. (D) Drug-resistant C94S mutant PITPNB also rescues the PI4P depletion. PITPNB-C94S that is mutated in the T58 position to decrease (T58A) or eliminate (T58E) PI binding shows impaired rescue. Grand average ± SEM from three independent experiments, each performed in triplicates. Source data are available online for this figure. (E) A cartoon showing the molecular targets of VT01454 and OSW1. (F) BRET analysis using the (2x)P4M PI4P sensor paired with Rab7-targeted Venus. Cells were pretreated with DMSO or VT01454 (100 nM) for 30 min before the addition of OSW1 (30 nM). BRET values were normalized to the control that received DMSO during preincubation as well as in place of OSW1. Data Information: Grand average ± SEM from three independent experiments performed in triplicates. The non-transfected DMSO (blue) and VT01454-treated (red) traces shown for reference are the same curves used in Fig. 1F, right panel. Only the PI-transfer competent PITPs can reverse the inhibition. Source data are available online for this figure.

**Table 1. Statistics of crystallographic data and refinement.**

| Crystal | PITPNA + VT01454 |
|---|---|
| PDB accession code | 8PQO |
| **Data collection and processing** | |
| Space group | P 2 2$_1$ 2$_1$ |
| Cell dimensions | |
| a, b, c (Å) | 50.33, 81.91, 94.70 |
| α, β, γ (°) | 90, 90, 90 |
| Resolution range (Å) | 25.42–2.30 (2.38–2.30) |
| No. of unique reflections | 17 995 (1760) |
| Completeness (%) | 95.55 (83.47) |
| Multiplicity | 13.7 (13.8) |
| Mean I/σ(I) | 21.5 (1.9) |
| R-merge | 0.127 (1.364) |
| R-meas | 0.131 (1.416) |
| CC$_{1/2}$ (%) | 99.9 (95.6) |
| **Structure solution and refinement** | |
| R-work (%) | 23.51 (33.78) |
| R-free (%) | 28.27 (44.87) |
| Ramachandran favored/outliers (%) | 98.38/0 |
| R.m.s.d. | |
| bonds (Å) | 0.005 |
| angles (°) | 0.74 |

Numbers in parentheses refer to the highest resolution shell.

dynamics simulations of PITPNA have also shown that in the cargo-free conformation, the designated lipid exchange loop penetrates into the membrane, and it was also suggested that the large movement of this short α2 helix during cargo binding reflects the lifting of the lipid cargo from the membrane and into the lipid binding cavity (Grabon et al, 2017; Schouten et al, 2002). These findings prompted us to further analyze the structural aspects that contribute to the membrane interactions of PITPs.

## Diacylglycerol promotes membrane association of the class I PITPs

Based on structural studies, it has been suggested that the "open", cargo-free structure of PITPNA represents the membrane-bound state

of the protein (Grabon et al, 2017; Schouten et al, 2002). Moreover, deletion of the C-terminal 11 amino acids in recombinant rat PITPNA was shown to strongly increase its membrane binding affinity, while also yielding a more relaxed conformation of the protein (Tremblay et al, 1998; Voziyan et al, 1996). This raises the question of whether the C-terminal truncation increases membrane binding via membrane penetration of the α2 helix, which would favor the open conformation of the PITP-fold. We compared the basal localizations of the full-length as well as the C-terminally truncated forms of PITPNA (PITPNA-Δ5), which were EGFP- or mRFP-tagged at their N-termini, and also monitored their response to AngII (100 nM) stimulation in HEK293-AT1 cells. Expressed wild-type PITPNA showed no discernible membrane localization and did not respond to AngII stimulation in any significant ways (YJK and TB, unpublished observations). In contrast, PITPNA-Δ5, while also mostly cytoplasmic, showed minor perinuclear membrane enrichment in resting cells. More importantly, this mutant showed prominent translocation to the PM after AngII stimulation (Fig. 5A,B). A similar localization pattern was observed with the PITPNB-Δ6 mutant, except that this construct also showed a prominent Golgi and some ER localization in the resting state, as well as a strong PM localization after AngII addition (Fig. 5C,D). The kinetics of this AngII-induced PM translocation were reminiscent of the DAG accumulation in the PM that has been documented in our previous studies (Kim et al, 2015). Co-localization of the PITPNA-Δ5 or PITPNB-Δ6 mutant with our DAG sensor [C1$_{ab}$ domain from mouse PKD (Kim et al, 2015)] was also prominent in the PM during AngII stimulation (Fig. EV2A,B). That the membrane levels of DAG are the primary determinants of the translocation of PITPs to the PM was further supported by the finding that inhibition of the conversion of DAG to PA in the PM by a DAG-kinase inhibitor during AngII stimulation further promoted the membrane association of the truncated forms of PITPNA and PITPNB (Fig. 5A–D). Accordingly, the DAG-mimic PMA also increased the PM localization of PITPNA-Δ5 (Fig. 5F).

## Structural features of class I PITPs contribute to their membrane binding

Given the DAG-driven localization of the truncated forms of PITPs, we aligned the sequences of PITPs with the DAG-binding C1 domains of several PKCs. We found that a proline-threonine-phenylalanine (PTF) sequence located in the α2 helical lipid-exchange loop, and conserved across all PITPs, showed a good alignment with a similar sequence found within the C1 domains of PKCs, which is located just ten residues upstream of an essential

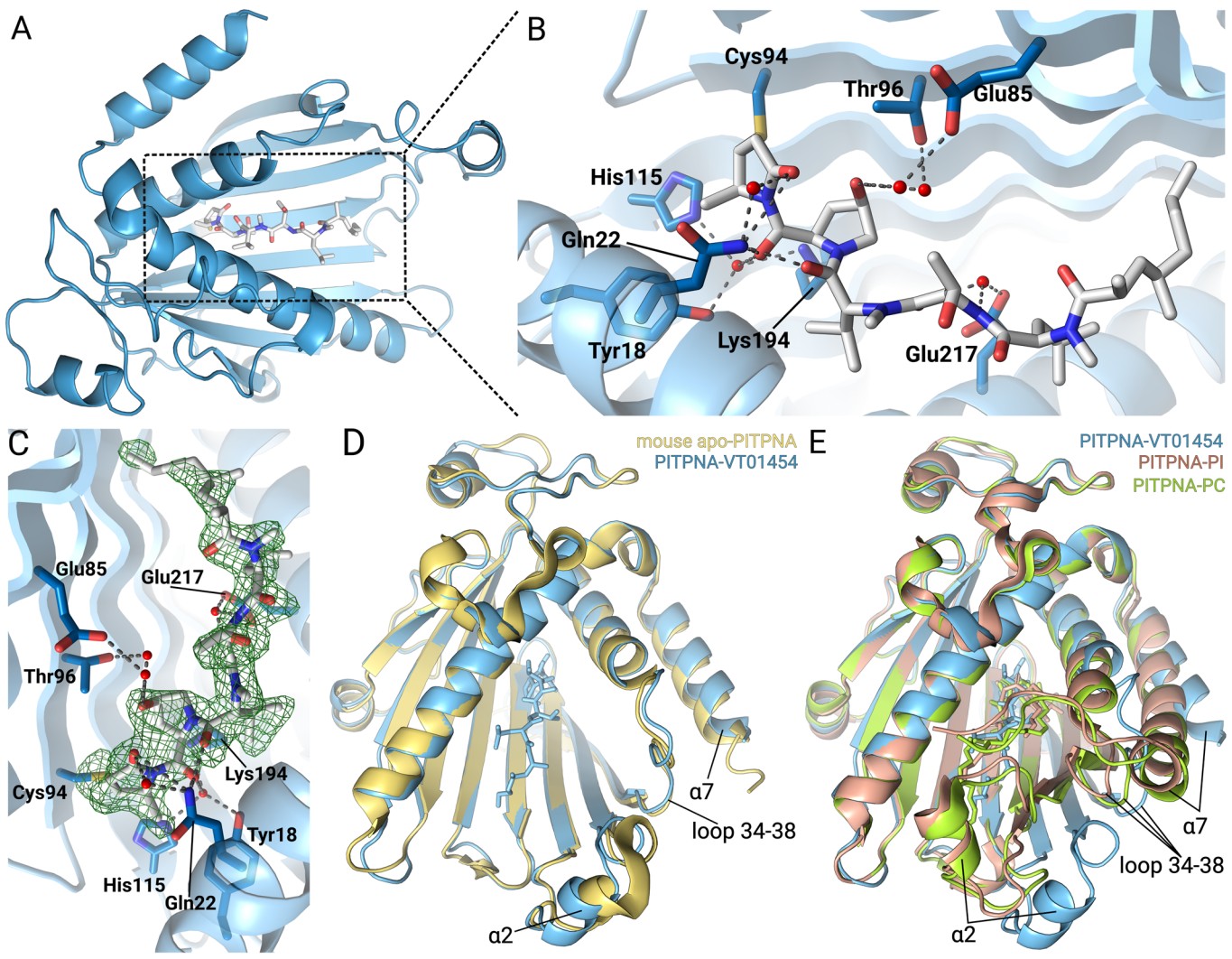

**Figure 4. Structural description of the PITPNA-VT01454 complex.**

(A) Structure of the PITPNA-VT01454 complex. VT01454 is shown in white in stick representation, while PITPNA is colored blue. (B) Detailed view of the VT01454 inhibitor in the lipid binding cavity. An identical color scheme is used as in panel (A). VT01454-interacting residues are represented as sticks. Water molecules engaged in the interaction are depicted as red spheres. Selected hydrogen bonds are depicted as gray dotted lines. (C) The electron density of VT01454 (white sticks) is shown in the simple Fo-Fc omit map contoured at 3D and colored dark green. (D) Structural alignment of mouse *apo*-PITPNA (light yellow, PDB ID:1KCM) and PITPNA-VT01454 (blue). Note that the inhibitor-bound protein assumes the open conformation. (E) Structural alignment of PITPNA-VT01454 (blue) and the PC- (light green, PDB ID:1T27) and PI-bound PITPNA (pink, PDB ID:1UW5) from rat and human, respectively. PyMol (Schrodinger, LLC) was used to create this Figure. Source data are available online for this figure.

membrane-oriented tryptophan residue (Fig. 5E, yellow) in the PKCs (see Fig. 5G for the positions of these residues in the respective structures). Residues at the +2 and +3 positions downstream from the conserved proline are highly hydrophobic in all PITPs including the class II members (Fig. 5E). Mutation of the two hydrophobic residues in this position to alanine (F71, V72 to AA) in the Δ5 form of PITPNA (or in the Δ6 form of PITPNB), completely eliminated the membrane binding in response to AngII stimulation (Fig. 5H, upper panels). These data together showed that C-terminal truncations promoted the membrane penetration of the α2 helix, and it does so in DAG-rich membrane domains. Two conserved tryptophane residues (W201, W202) previously identified (Tilley et al, 2004) were also required for membrane

localization, as their mutation to alanine also completely eliminated the AngII-induced membrane translocation of the C-terminally truncated PITPs (Fig. 5H, lower panels).

After identifying the residues that are critical for membrane interaction, it was of interest to perform rescue experiments using PITPNB variants with mutations in the membrane interface prepared in the C94S background. Deletion of the last 6 residues in PITPNB, which compromises the retention of cargo in PITPs, greatly reduced but did not eliminate the ability of the protein to maintain PM levels of PI4P following VT01454 treatment (Fig. 6A). In contrast, mutation of the double Trp residues (W201A, W202A), completely prevented rescue. In contrast, mutation of the FV residues (F71, V72 to AA) within the lipid exchange loop of the

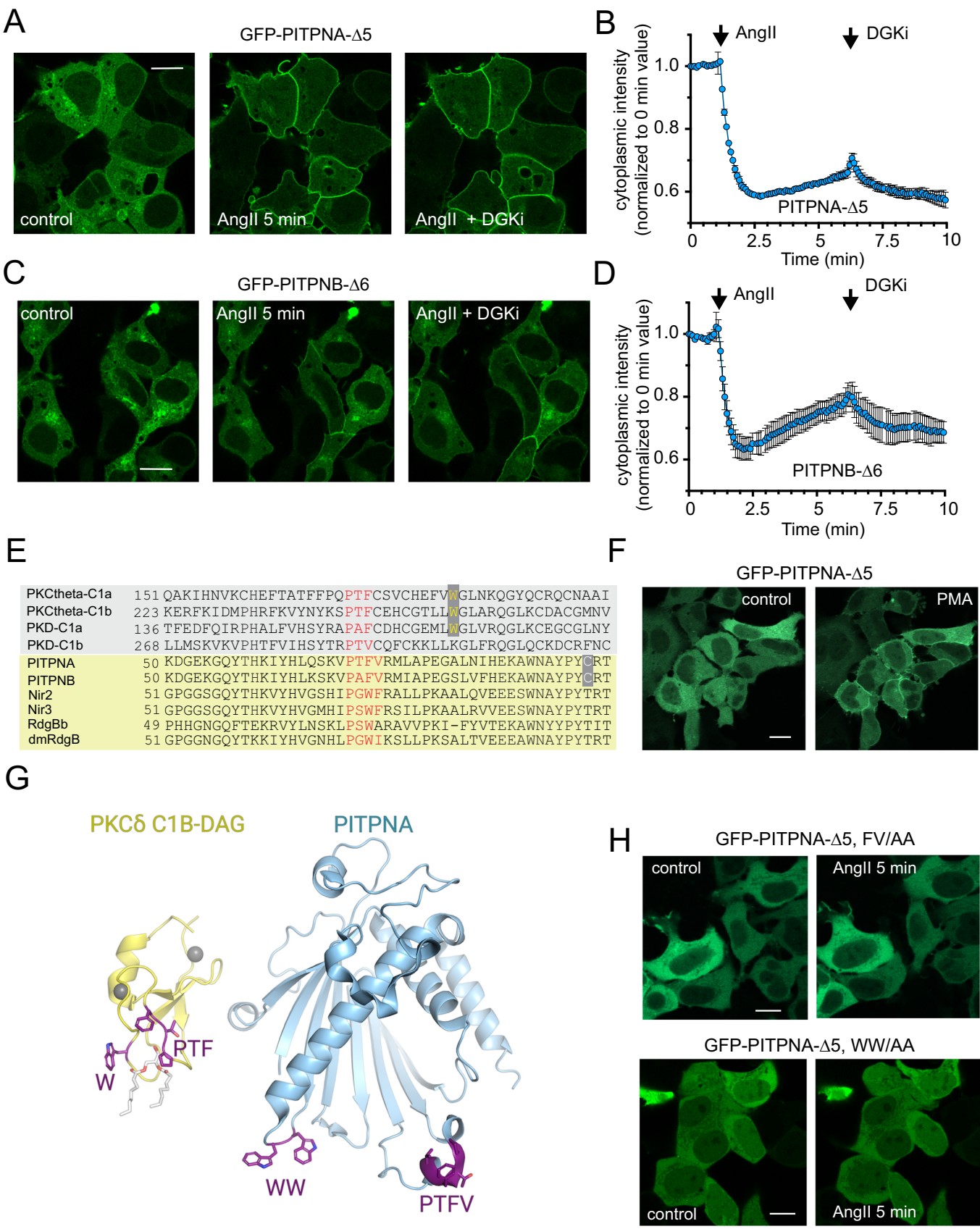

**Figure 5.** **C-terminal truncated PITPs recognize DAG-rich membranes.**

(**A**) Representative images showing the localization of EGFP-tagged PITPNA truncated at the five C-terminal amino acids (EGFP-PITPNA-Δ5) expressed in HEK293-AT1 cells. AngII (100 nM) stimulation caused rapid association of the protein with the PM also causing a drop in the cytoplasmic EGFP intensity (middle panel). The addition of a DAG kinase inhibitor, R59022 (50 μM) causes a further increase in membrane localization (right panel). (**B**) The cytoplasmic intensity of EGFP-PITPNA-Δ5 was quantified in regions of interest outside the nucleus. Translocation to the PM is reflected in a decrease in cytoplasmic fluorescence. The Intensity values were normalized to the 0 min values and expressed as ratio changes over time Grand averages ± SEM are shown from four independent experiments (8–18 cells monitored in each). (**C, D**) Same as panels A,B using EGFP-PITPNB-Δ6. Note the more prominent Golgi and some ER localization even before AngII stimulation. (**E**) Sequence alignment of C1 domains from PKC and PKD and the PITP domains of class I and class II PITPs. Note the conserved "P-T/A-F/V" sequence (red) in the two groups of molecules. Also, note that the double Trp residues (W201,202) of PITPs is not covered in this alignment. Their functional role is analogous to that of the conserved Trp (yellow on gray background) found in C1 domains. Also shown is the VT01454-reactive C94 residue (white on a gray background) that is not conserved in the class II proteins. (**F**) Cellular distribution of EGFP-PITPNA-Δ5 was monitored after PKC activation after PMA (100 nM) treatment. Note the translocation of the protein to the PM. (**G**) Structural similarities between the PKC C1 domain and PITPs in their hydrophobic interactions with the membrane. PDB IDs: 7L92 and 1KCM for PKCδ C1b and PITPNA, respectively. (**H**) Membrane association of EGFP-PITPNA-Δ5 after AngII stimulation was eliminated when the two hydrophobic residues (F71, V72) localized in the lipid exchange loop or the double Trp (W201, W202) were mutated to alanine. Scale bar: 10 μm. Source data are available online for this figure.

full-length PITPNB, did not impair its ability to reverse the effects of VT01454 on the PM levels of PI4P (Fig. 6A). While these latter data were surprising, they were in agreement with a previous report regarding the reduced functionality of the C-terminally truncated PITPNA (Hara et al, 1997) but not the F72A mutant PITPNA protein in an assay based on reconstituting PLC activity in permeabilized cells (Tilley et al, 2004).

## VT01454 facilitates the interaction of class I PITPs with membranes

Given that VT01454 blocks cargo binding, and the solved structure suggests that it stabilizes the "open" conformation of PITPs (see Figs. 4D, 5G), which, in turn, enhances membrane binding, we next examined how VT01454 affects the localization of the wild-type PITPNA and PITPNB proteins. The addition of VT01454 to cells expressing the mRFP-tagged PITPs showed a minor effect, with some cells showing weak ER localization of the PITPs. However, after AngII stimulation, the VT01454-bound PITPs showed a clear association with both the PM and the ER, although this latter may only become more obvious as the cytoplasmic signal decreases. The overall membrane association appeared to be stronger in the case of PITPNB (Fig. 6B) and also required the two surface Trp residues (W201, W202, Fig. 6C). Importantly, mutation of the Cys94 residue that is critical to react with VT01454 to serine (C94S), completely prevented the VT01454-induced membrane localization as well as the AngII-induced translocation of the otherwise wild-type PITPNA protein to the membranes (Fig. 6D).

Taken together, these data showed that VT01454 can promote stabilization of the open conformation of class I PITP proteins, facilitating their interactions with the membrane. This increased PITP-membrane interaction was not, however, as large as those seen after C-terminal truncations were performed. Surprisingly, however, despite their presumably impaired ability to close in on their lipid cargo, C-terminally truncated PITPs still remain partially active. In contrast, the hydrophobic F71, V72 residues in the lipid-exchange loop that confer DAG recognition appear to be dispensable, at least under our overexpression assay conditions.

## Complementary roles for class I and class II PITPs in supporting PI(4,5)P$_2$ maintenance during PLC activation

It has been previously shown that the class II PITP, Nir2, transports PI from the ER to the PM (Chang and Liou, 2015, 2016; Kim et al,

2013; Kim et al, 2022) in exchange for PA, which is returned to the ER from the PM (Kim et al, 2015; Yadav et al, 2015). This process was found to be important for the maintenance of the signaling pool of PI(4,5)P$_2$ in the PM when PLC was activated (Fig. 7A). Given the complete lack of effect of the VT01454 compound on resting PI(4,5)P$_2$ levels (Fig. 1E), we wanted to assess the extent to which class I PITPs contribute to the replenishment of the PI(4,5)P$_2$ levels in the PM during strong PLC activation. For this, we used BRET analysis and challenged the cells with the Gq-coupled muscarinic, M1 receptor agonist, carbachol, and followed the recovery kinetics of PI(4,5)P$_2$ after terminating the response by adding the antagonist, atropine. For comparison, we also used the PI4KA inhibitor, GSK-A1 (100 nM), which, strongly inhibits the recovery phase of PI(4,5)P$_2$ replenishment upon terminating PLC activation without decreasing the resting PI(4,5)P$_2$ levels (Hammond et al, 2012; Toth et al, 2016). As shown in Fig. 7B, pretreatment of the cells with VT01454 (100 nM) did not affect resting PI(4,5)P$_2$ levels, and had only a minor effect on the PI(4,5)P$_2$ recovery rate, which contrasted the sizable inhibitory effects of GSK-A1 (Fig. 7B). We noted that both inhibitors reduced the relative size of PI(4,5)P$_2$ drop, and the reason for this change is still under investigation. Regardless, the finding that PI4KA inhibition had a larger inhibitory effect on PI(4,5)P$_2$ recovery following carbachol stimulation than VT01454 treatment, suggested that PI4KA in the PM receives PI from a source other than the class I PITPs. Obvious candidates for such a PI delivery role are the class II PITPs, including the ubiquitously expressed Nir2. To test this possibility, we compared Nir2 K/O cells with parental HEK293-AT1 cells and analyzed the effects of VT01454 and GSK-A1 treatment on the level and recovery rate of PI(4,5)P$_2$ following the carbachol-atropine treatment regime. Pretreatment of Nir2 K/O cells with VT01454 reduced the rate of PI(4,5)P$_2$ recovery after atropine almost to the same level observed in cells pretreated with GSK-A1 (Fig. 7C). This suggested that class I PITPs made a significant contribution to PI delivery to the PM for PI(4,5)P$_2$ synthesis when Nir2 was eliminated.

These findings prompted us to compare the PI4P and PA changes in parental and Nir2 K/O cells in response to inhibition by VT01454 or GSK-A1 using BRET analyses. As shown in Fig. 7D, pretreatment of parental cells with VT01454 (100 nM) for 30 min, reduced PI4P levels to a very low level that was comparable to what was found after GSK-A1 (100 nM) treatment (also see Fig. 1D). Accordingly, carbachol stimulation caused only a small additional decrease, and the recovery of PI4P levels after atropine addition was

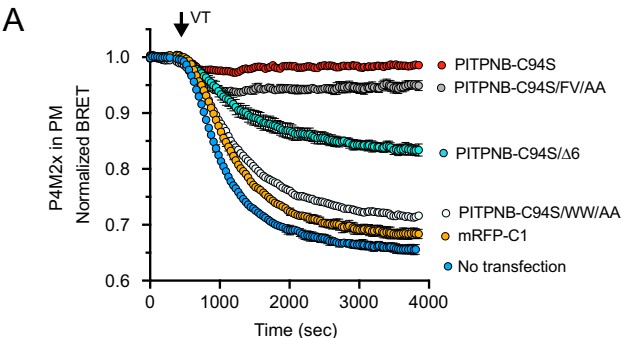

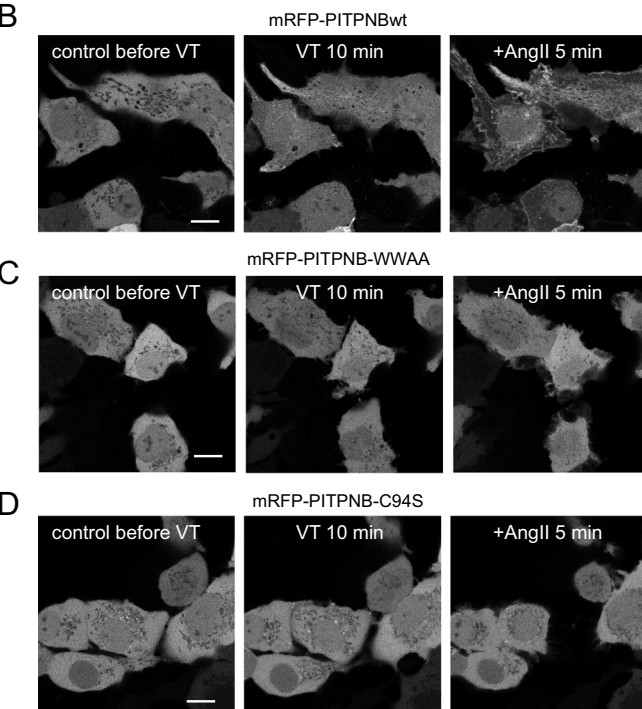

**Figure 6. PITPs treated with VT01454 associate with membranes.**

(A) Rescue experiments with PITPNB variants with mutations that affect their membrane interaction. For BRET analyses, see Legend to Fig. 4. Note that the two Trp residues are essential for rescue, and the C-terminally truncated protein is still somewhat functional. In contrast, in the context of the full-length protein, the mutated FV residues only slightly impair the ability of the protein to rescue. Grand average ± SEM from for independent experiments performed in triplicates. Source data are available online for this figure. Also note that the VT-treatment (blue), PITPNB-C94S (red), and mRFP-C1 rescue (yellow) curves are shown for reference, which are the same curves that are shown in the respective panels in Fig. 3. Scale bars: 10 μm. (B) mRFP-tagged wild-type PITPNB was expressed in HEK293-AT1 cells and treated with VT01454 (100 nM) for 10 min (middle), followed by AngII (100 nM) stimulation (right panel). VT01454 treatment caused a modest enrichment of the protein in membranes, whereas AngII stimulation increased the localization of PITPNB primarily in the PM (right panel). (C) Mutation of W201 and W202 to alanine in PITPNB abolished their membrane binding after VT01454 and AngII treatment. (D) VT01454-resistant C94S mutant of PITPNB doesn't bind to the membrane after VT01454 and AngII treatments. Source data are available online for this figure.

small but still reached the pre-stimulatory low levels in VT01454-, but not in GSK-A1-pretreated cells. Nir2 K/O cells showed a slightly slower recovery of PI4P from the carbachol-induced decrease compared to the parental cells (Fig. 7E), and both VT014154 and GSK-A1 pretreatment yielded comparably low PI4P levels, which were not appreciably changed further by either carbachol or atropine treatment (Fig. 7E).

Since PLC activation also results in the production and accumulation of PA in the PM [see e.g. (Kim et al, 2022)], limiting the supply of PI is expected to impact the amount of PA that accumulates in the PM during stimulation of PLC. In spite of its small effect on PI(4,5)P$_2$ recovery, in parental HEK293-AT1 cells, VT01454 treatment had a major impact on the basal level of PA and its accumulation at the PM during carbachol treatment (Fig. 7F). However, GSK-A1 treatment still had a stronger inhibition in these parental cells (Fig. 7F). Nir2 K/O cells showed a somewhat reduced PA response to carbachol stimulation and a slower rate of PA clearance from the PM after the addition of atropine (Fig. 7G). The slower PA clearance after atropine treatment in Nir2 K/O cells was consistent with the reported role of Nir2 in recycling PA from the PM to the ER for use in PI resynthesis (Kim et al, 2015). Again, the difference between the inhibitory effects of VT01454 and GSAK-A1 on the accumulation of PA in the PM was reduced in Nir2 K/O cells compared to that of parental cells, although GSK-A1 was still more effective than VT01454 in the Nir2 K/O cells (Fig. 7F,G).

Collectively, these data showed that class I and class II PITPs are both involved in the supply of PI to the PM for PI4P and PI(4,5)P$_2$ generation during the high demand following PLC activation. Notably, however, even though the overall flux through PI4P and PI(4,5)P$_2$ during PLC activation is greatly reduced in cells treated with VT01454, which is best judged by the changes to PA accumulation in the PM, the restoration of PI(4,5)P$_2$ levels following stimulation is more efficiently supported by the class II PITPs that are specifically recruited to ER-PM contact sites during PLC activation.

## Inhibition of class I PITPs affects several other lipid classes beyond PPIns

Next, we performed lipidomic analyses on HEK293-AT1 cells treated with VT01454 (100 nM) for 90 min. While most of our BRET and confocal experiments covered shorter treatment times, we chose 90 min treatment for this measurement to ensure that with longer times the changes are larger and, hence can be evaluated with more certainty. This analysis showed the largest changes in PA, but strikingly, increases only in the levels of unsaturated PA species, including the mono-unsaturated 32:1 and 34:1 forms, and, to a smaller extent, di-unsaturated 34:2 molecular forms, while the saturated PA species remained unchanged (Fig. 8A). A moderate decrease was observed in DAG, which was restricted only to two short chain-length and saturated DAG species, namely 30:0 and 32:0 (Fig. 7B). Reduced levels were observed in several species of PS, the largest reduction being in the 34:1 and 36:1 forms (Fig. 8D) as well as a slight elevation in the 38:4 form of PI (Fig. 8C). Cholesteryl esters were generally increased while several forms of TAG showed a slight decrease (Fig. 8E,F). No significant changes were observed in the levels of PC,

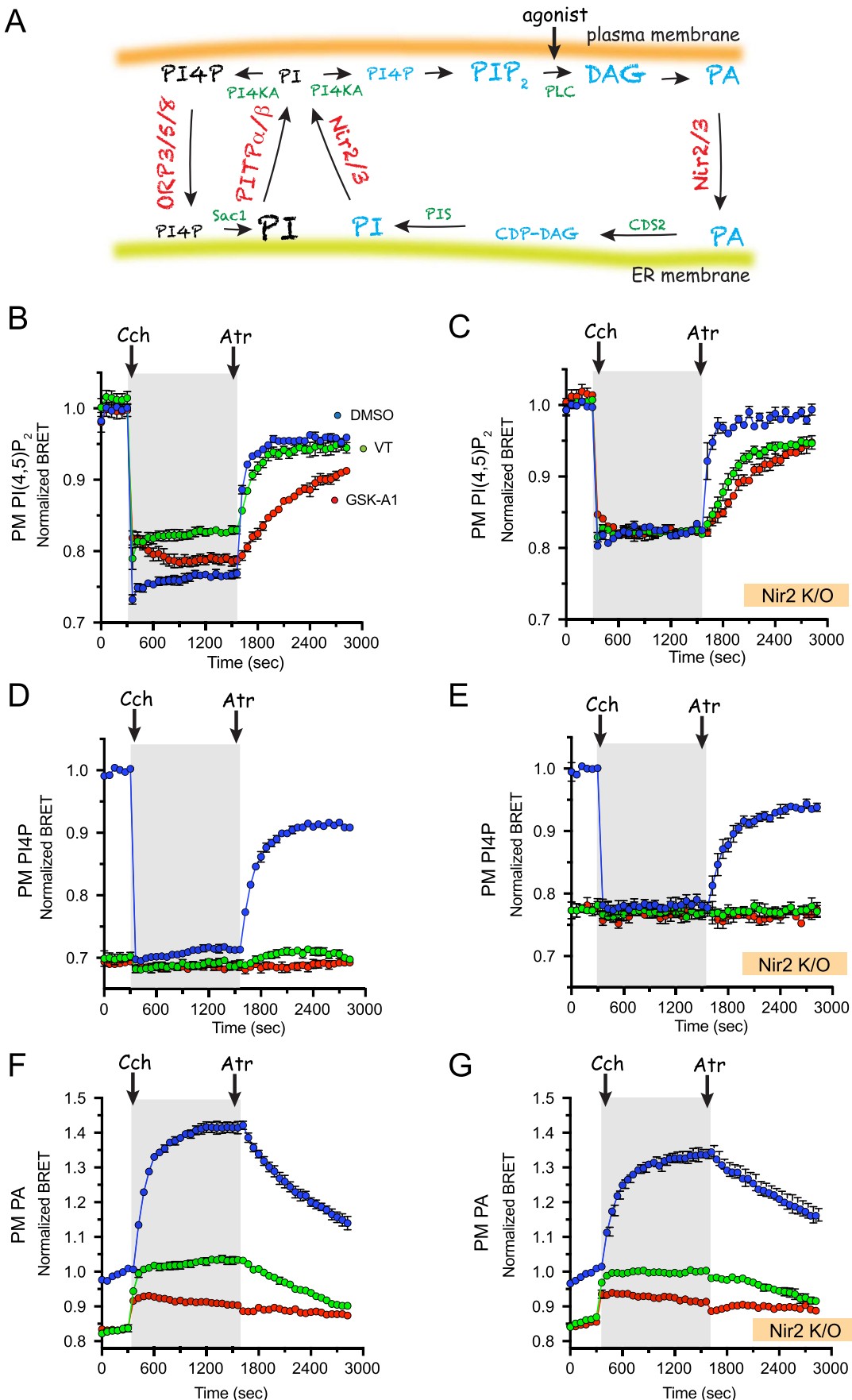

**Figure 7. Contribution of class I and class II PITPs for PI delivery to the PM during PLC activation.**

(A) Cartoon describing the components of the "PI-cycle". Agonists-coupled PLC activation stimulates $PI(4,5)P_2$ hydrolysis with the generation of inositol 1,4,5-trisphosphate and DAG, the latter is rapidly converted to PA in the PM. Abbreviation: PIS (PI synthase), CDS (CDP-DG synthase), Sac1 (PPIn phosphatase). PI is rapidly resynthesized in the ER from PA, but for this to happen, PA needs to be transferred to the ER. Newly synthesized PI also has to be delivered to the PM to maintain the PM pool of $PI(4,5)P_2$. The class II PITP, Nir2 has been shown to perform the PA/PI exchange in ER-PM contact sites during PLC activation (see text for original References). In parallel pathways, PI4P generated from PI in the PM by the PI4KA enzyme is used to facilitate the transport of PS from the ER to the PM by the ORP5/8 proteins. PI supply is also critical for the efficient operation of this transport pathway. (B–D) BRET analysis following the PM levels of $PI(4,5)P_2$ (B, C), PI4P (D, E), and PA (F, G) using the (2x) P4M, PLCδ1-PH and NES-Spo20, respectively, as LBDs, paired with PM-targeted mVenus. Cells were transfected for 24–28 h with the indicated BRET construct together with the M1 muscarinic receptor. Carbachol (Cch, 100 µM) was added to stimulate PLC and atropine (Atr, 10 µM) was added to terminate the response (gray areas show the duration of the stimulation). Cells were pretreated with DMSO, GSK-A1 (100 nM), or VT01454 (100 nM) for 30 min before the start of BRET measurements. HEK293-AT1 parental cells (B, D, F) or their Nir2 K/O version (C, E, G) are shown. Data Information: Grand averages ± SEM from three experiments are shown, each performed in triplicates. Source data are available online for this figure.

phosphatidylethanolamine (PE), phosphatidylglycerol (PG), or sphyngomyelin (SM) (Fig. EV3A–D).

To understand the connection between PITP inhibition and the changes observed in lipidomic analysis, we performed further metabolic labeling experiments. Our previous study showed that decreases in the PM levels of PI4P induced by PI4KA inhibition had a strong and immediate impact on PS biosynthesis (Sohn et al, 2016). Such decreases in PI4P in the PM could be responsible for the observed drop in PS levels in VT01454-treated cells. Therefore, we also investigated whether inhibition of class I PITPs also affected the PS synthetic rate using isotope labeling experiments with [$^{14}$C]-serine. Treatment of cells with VT01454 (100 nM) strongly inhibited the labeling of both PS and PE in a 2 h incubation, which was similar in magnitude to the effect of acute PI4KA inhibition (Sohn et al, 2016) (Fig. EV2C). These data showed that PITPs are important for the maintenance of the PI4P gradient that is established between the PM and the ER, which supports the transport of PS out from the ER. Together with the strong feedback inhibition exerted by PS on the PS-synthesizing enzyme, PSS1, a defect in PS export from the ER can directly cause a significant inhibition of PS synthesis (Sohn et al, 2016).

### Acute inhibition of class I PITPs inhibits PI synthesis

The PA species found to accumulate in the lipidomic analysis of VT01454-treated cells suggested that PI synthesis might be stalled under conditions when PI is not extracted from the ER, leading to the accumulation of its precursor, PA. In fact, the PI synthase enzyme is known to work in the reverse mode and catalyze robust "inositol exchange activity" (Kim et al, 2022; Lykidis et al, 1997). Therefore, we tested the effects of VT01454 on the incorporation rate of *myo*-[$^3$H]inositol into PI in a 1 h incubation both in control cells and in cells stimulated by AngII. Note that these relatively short inositol-labeling experiments primarily monitor the PI synthetic rate, whereas the longer, close-to equilibrium-labeling regime that was used in the experiments described earlier and shown in Fig. 1A, reflects on the total level of PI, which is not changed during the period of VT01454 treatment used in these experiments. As shown in Fig. EV2D, knockout of either PITPNA or PITPNB, had no effect on the basal PI synthetic rate, nor did they show any effect on the AngII-induced increase in PI synthesis, which was almost tenfold in all cases. However, in the presence of VT01454 (100 nM) the incorporation rate was reduced by more than 50%. These data were consistent with a reduced rate of PI synthesis in cells where PI transport from the ER by class I PITPs is inhibited.

## Discussion

Extensive studies on mammalian class I PITPs and their functional yeast homolog, Sec14, led to the general conclusion, that these proteins deliver PI from the ER to the other organelle membranes. This includes essential roles in supporting $PI(4,5)P_2$ production in the PM to maintain PLC-mediated signaling, and PI4P production in the Golgi complex to serve the multiple functions within this organelle (Ashlin et al, 2021; Mousley et al, 2012). It has been an important and highly debated question whether PITPs simply transport PI between membranes or serve as instructive context-specific regulators of PI kinases (Cockcroft and Garner, 2011; Grabon et al, 2019).

Our studies are the first that were able to investigate the role of class I PITPs in intact cells using a comprehensive pharmacological approach, allowing acute and simultaneous inactivation of both class I PITPs, PITPNA, and PITPNB, which was not possible to accomplish with previously used genetic means. Our analysis showed that acute inhibition of class I PITPs did affect PI4P formation in the PM and Rab7-positive endosomal compartments, both of which show low steady-state PI levels (Pemberton et al, 2020; Zewe et al, 2020). However, treatment with VT01454 had a much smaller effect on PI3P generation in Rab5-positive early endosomes and failed to acutely reduce PI4P levels in the Golgi compartment. Since PI3P in the Rab5-positive compartment was still largely dependent on the activity of the class III PI3K, Vps34, we must assume that this compartment receives PI by a mechanism that is less dependent on class I PITPs. This question needs further investigation and is currently being pursued in our laboratory. The high level of PI that is detected in the Golgi complex (Pemberton et al, 2020; Zewe et al, 2020) is the most likely reason why the Golgi compartment can sustain PI4P production for an extended period even when the PITPs are inhibited. While the situation might be very different when the PITP proteins are genetically inactivated, previous studies also showed minimal or no changes in the Golgi level of PI4P when the individual PITPs were removed or downregulated by genetic means (Alb et al, 2002; Cockcroft and Carvou, 2007).

Another long-standing question that we were able to address here was the extent of contributions by class I PITPs to the supply of PI to the PM during PLC-mediated consumption of $PI(4,5)P_2$. Our results showed no effect VT01454 on resting $PI(4,5)P_2$ levels in the PM, even though the PM levels of PI4P were drastically reduced. This was similar to what was already observed when

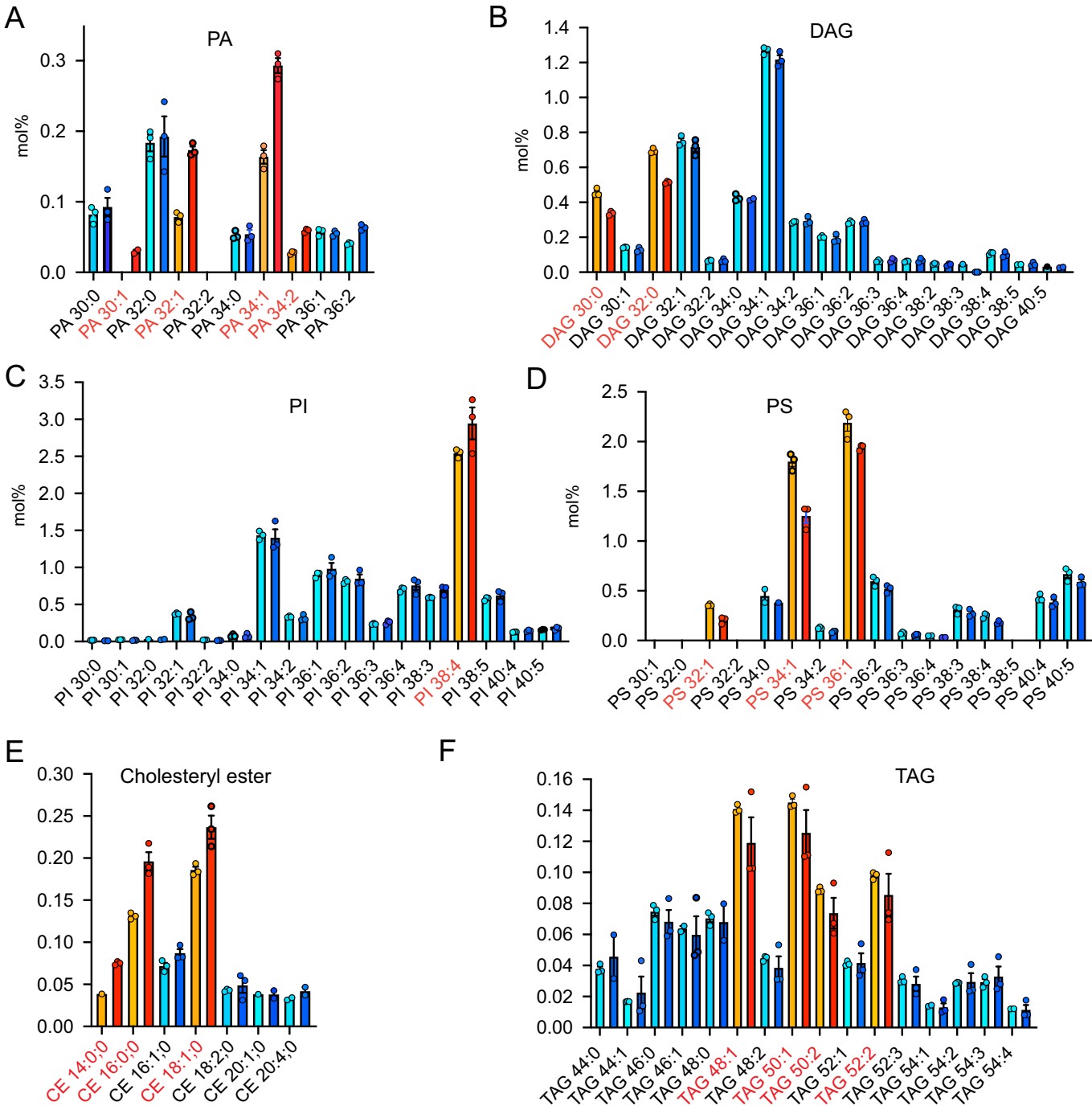

**Figure 8. Lipidomic analyses of HEK293-AT1 cells treated with VT01454.**

HEK293-AT1 cells were treated with VT01454 (100 nM) or DMSO for 90 min and prepared for lipidomic analyses, which were performed by Lipotype as described in the Methods Dark columns show VT01454-treated cells. (A) Phosphatidic acid; (B) Diacylglycerol; (C) Phosphatidylinositol; (D) Phosphatidylserine; (E) Cholesteryl ester; (F) Triacylglycerol; Blue colors designate lipid species that show minor or no change, whereas orange/red columns highlight lipid species showing the largest changes in response to the inhibitor treatments. Note that among the PA species, only the unsaturated forms show increases in response to VT01454 treatment. (E, F) Increases in cholesteryl esters and decreases/increases in selected TAG species were more broadly distributed among the various fatty acyl side chain forms. Source data are available online for this figure. Data Information: Means ± SEM and the individual data points are shown from three independent biological samples from one experiment that was repeated with essentially the same results. Source data are available online for this figure.

inhibiting PI4KA with the specific inhibitor GSK-A1 [e.g. (Bojjireddy et al, 2014; Gulyas et al, 2022)], a puzzling observation that suggests that the PM levels of PI(4,5)P$_2$ can be maintained at a wide range of PI4P concentrations as long as the PI(4,5)P$_2$-consuming PLC enzymes are not highly active. Importantly, when PLCs are stimulated, recovery of the PM levels of PI(4,5)P$_2$ were not significantly impaired by VT01454 pretreatment despite a substantially reduced flux through the PPIn pools in the PM. Only in cells lacking the class II PITP, Nir2, did the contribution of class I PITPs become critical for PI(4,5)P$_2$ resynthesis. These findings suggests that class I PITPs may primarily support the maintenance of the PI4P-driven non-vesicular transport pathways between the PM and the ER rather than supply the PI(4,5)P$_2$ synthetic machinery, which seems to rely upon the PI and PA exchange functions of the class II Nir2/Nir3 PITPs. Our pharmacological studies could not tell which of the two class I PITPs are more important for these functions. Reports on platelets deficient in PITPNA (Zhao et al, 2017) or PITPNB (Zhao et al, 2023) suggested differences between the two PITPs in supporting various angles of platelet functions. However, our studies in the HEK293 cell background were not able to detect a major loss of PI4P or PI(4,5)P$_2$ levels in single PITPNA or PITPNB K/O cells [see Fig. 1A and (Kim et al, 2022)].

We devoted a significant effort to determine the mechanism of membrane interaction of the class I PITPs. C-terminal truncations were previously reported to significantly enhance the interaction of recombinant rat PITPNA with lipid vesicles and reduced their PI or PC transfer activity by ~60% in vitro (Tremblay et al, 1998). Using intact cells, we found that both proteins truncated in their last five (PITPNAΔ5) or six (PITPNBΔ6) C-terminal residues showed prominent binding to DAG-rich membranes, which required the hydrophobic residues within the conserved PTFV (PITPNA) or PAFV (PITPNB) sequence, which is also found in the DAG-sensing C1 domains of PKCs (Fig. 5E). This sequence is located within the α2 helical "lipid exchange loop" of the protein that dips into the membrane interface in the open molecular state (Fig. 5G) and shows the highest conformational change upon binding to lipid cargoes (Grabon et al, 2017; Schouten et al, 2002; Tilley et al, 2004; Yoder et al, 2001). Based on these findings, it is tempting to speculate that DAG-rich membranes could facilitate lipid exchange in the cargo binding avity of class I PITPs. Since activation of PLC in the PM consumes PI4P and PI(4,5)P$_2$, while also generating DAG, this mechanism could ensure delivery of PI to the PM under higher demands.

The reliance of the VT01454 inhibition on covalent interaction with the Cys94 residue at the base of the lipid binding cavity, allowed us to analyze features of PITPs that are critical to their function in rescue experiments using C94S mutant variants. These experiments are of great interest given the extensive mutational studies reported in the literature assessing the functions of class I PITPs in liposome experiments and permeabilized cells [see (Ashlin et al, 2021) for original citations]. C94S mutations made both PITPNA and PITPNB resistant to VT01454, and while it had some effect on the ability of PITPNA to extract PC from membranes, we found that these mutants were functionally fully competent at least for the processes that we have examined in live cells. These rescue studies showed that PI-binding was essential for their functions in supporting PI4P formation in the PM and Rab7-positive endosomes. The rescue experiments were also instructive regarding the

importance of membrane interaction of these proteins in intact cells. We confirmed the critical importance of the tandem Trp residues (W201, W202) that were previously described as affecting the membrane interaction and lipid transfer of PITPs in vitro and in permeabilized cells (Phillips et al, 2006; Tilley et al, 2004). The short C-terminal truncation of PITPNB (Δ6) in the C94S background reduced the activity of the protein by about 50%, whereas mutation of the FV residues to AA in the lipid exchange loop in the full-length PITPNB-C94S had only a minor effect (Hara et al, 1997). All of our rescue data fit very well with the in vitro behavior of PITPs described earlier by the Cockcroft group (Hara et al, 1997; Tilley et al, 2004). Importantly, treatment of the cells with VT01454 caused a slow association of the PITPNA and PITPNB proteins with both the PM and the ER, which was rapidly enhanced after PLC activation. This is consistent with the structural description of the VT01454-bound PITPNA, which shows the inhibitor stabilizing the open conformation that, in turn, has a higher membrane affinity. This finding also suggests that PLC activation increases cargo exchange in the PITPs, most likely through the generation of DAG, which would help to facilitate membrane binding of the inhibitor. It is notable that the reactivity of the C94 residue with the thiol-reactive reagents was found to be facilitated by membrane binding (Shadan et al, 2008; Tremblay et al, 2001). The VT01454-induced membrane localization, however, was less prominent than that of the C-terminally truncated proteins. The C-terminus of the protein [also termed the "lid" (Shadan et al, 2008)] likely acts as a latch that traps the ligand in the binding pocket through stabilization of the long C-terminal helix in the closed conformation. The fact that the C-terminal tail of the protein is not resolved in the open structure [(Schouten et al, 2002) and this study] suggests that additional interactions with the membrane are likely to control its movements.

Lastly, our lipidomics analyses of cells treated with VT01454 showed changes in several lipid classes without any major change in the absolute levels of PI. The most prominent of these changes was the selective accumulation of PA, which was remarkably restricted to only the unsaturated acyl-chain forms. In contrast, moderate decreases in DAG levels were observed that only affected the short, more saturated acyl-chain forms. The decreased PS levels were consistent with previous studies showing that reducing levels of PI4P in the PM results in the rapid inhibition of the PSS1 enzyme due to the strong product inhibition on PSS1 that is exerted by the PS that is retained in the ER as a result of the diminished PI4P gradient that no longer exists between the PM and the ER (Sohn et al, 2016). This was directly confirmed by isotope flux studies using [$^{14}$C]-serine labeling. Similarly, we attributed the accumulation of PA to the impaired exit of the synthesized PI from the ER, which, together with the reversibility of the PI synthesizing machinery, could cause the accumulation of the metabolic precursor, PA. The reduced myo-[$^3$H]inositol labeling of PI was also consistent with an inhibition of the conversion of PA to PI. The fact that only the unsaturated PA species accumulates also supports the idea that there is a functional compartmentalization of the PA pools that are directed towards various metabolic fates. Also, the reduced levels of DAG and TAG, as well as the accumulation of some forms of cholesteryl esters, suggest that these additional metabolic changes also influence neutral lipid storage in the ER. While some of these lipid changes can be traced back to the inhibition of PI transport by class I PITPs, others, such

as those in TAG and cholesteryl esters, could be secondary or possible off-target effects. Addressing these possibilities will require further studies.

It is of particular interest that VT01454 was identified as part of a screen for inhibitors of the Hippo signaling pathway (Li et al, 2022), and the most profound inhibitory effects of VT01454 were on the PI4P levels in the PM. The rapid reductions in PM levels of PI4P are also observed with the PI4KA inhibitor, GSK-A1. PI4KA has already been implicated in the control of the Hippo pathway in *Drosophila* (Tan et al, 2014; Yan et al, 2011), and $PI(4,5)P_2$ is believed to be the primary acidic lipid required for anchoring Merlin to the PM and assembling the active LATS1/2 complex that keeps the Hippo pathway off (Chinthalapudi et al, 2018; Hong et al, 2020; Mani et al, 2011). However, unless PLC was strongly activated, neither VT01454 nor GSK-A1 treatment significantly dropped the $PI(4,5)P_2$ levels in the PM, which raises the possibility that the Hippo pathway activity may depend on the PM levels of PI4P and the tightly co-regulated anionic PS as much as it does on the total amounts of $PI(4,5)P_2$.

In summary, our studies present the first comprehensive analysis of the role of class I PITPs in cellular lipid metabolism using a pharmacological approach in intact cells. The structure of PITPNA with the bound to the VT01454 inhibitor reveals an open conformation that is also associated with increased membrane binding. While class I PITPs supply several organelles with PI for sustained production of PPIns, their acute inhibition reveals important effects on several additional lipid classes and highlights the central role of these non-vesicular lipid transfer proteins for orchestrating cellular lipid metabolism.

# Methods

## Reagents

Angiotensin II (human octapeptide) was from Bachem (Vista, CA). Atropine and DAG kinase inhibitor I (R59022) were purchased from Sigma Aldrich (St Louis, MO). Coelenterazine h (1-361301-200) was purchased from Regis Technologies (Morton Grove, IL). GSK-A1 inhibitor was described previously (Bojjireddy et al, 2014). VPS34-IN-1 was obtained from Selleckchem (Houston, TX). VT01454 has been described previously (Li et al, 2022). MI-14 was a kind gift from Dr. Radim Nencka (Institute of Organic Chemistry and Biochemistry AS CR, Prague 6, Czech Republic), and OSW1 was a generous gift from Dr. Matthew Shair (Department of Chemistry and Chemical Biology, Harvard University). [$^{14}$C]-acetic acid and [$^{14}$C]-serine (SA: 52 and 55 mCi/mmol, respectively) were purchased from American Radiolabeled Chemicals (St Louis, MO), and *myo*-[$^3$H]inositol (SA: 98 Ci/mmol) was from PerkinElmer (Waltham, MA). The polyclonal rabbit anti-PITPNA (clone "103") and monoclonal anti-PITPNB (clone "1C1") antibodies (Carvou et al, 2010) were kind gifts of Dr. Shamshad Cockcroft (University College, London, UK). All other reagents were of the highest molecular biology grade.

## DNA constructs

The human PITPNA was cloned from human brain cDNA (Clontech) using PCR amplification and subcloned into pEGFP-

C1 (all of PCR primers are listed in Table EV1). The human PITPNB (accession number BC018704, clone number 4643509 purchased from Open Biosystems) was also subcloned into pEGFP-C1, and a ten amino acid residue linker (GGAGGAAAGA) was inserted between the EGFP tag and the PITPNB protein (synthesized as an oligomer pair). Truncated mutations (Δ5 or Δ6) of PITPs were generated by side-directed mutagenesis by introducing of stop codon using the QuikChange mutagenesis kit from Promega. PITP constructs were also subcloned into mRFP plasmids using simple restriction digestions to replace EGFP with mRFP. PITPNA was also subcloned into pET-19b plasmids for generating recombinant proteins (see below). Human PITPNA and PITPNB numbering are used throughout the manuscript. The corresponding residues in the rat and mouse sequences used in most studies in the literature are one number higher; therefore, Cys94 of human PITPs corresponds to Cys95 in the rat or mouse proteins.

The single plasmid-based BRET constructs to monitor PI, PI4P, $PI(4,5)P_2$, and PA in the PM, PI3P in the Rab5 compartment, and PI4P in the Rab7 compartment have been described previously (Baba et al, 2019; Kim et al, 2022; Pemberton et al, 2020). EGFP-FAPP1-PH and EGFP-CERT-PH were described in our previous study (Toth et al, 2006), and EGFP-FAPP2-PH was kindly provided by Dr. Maria Antonietta De Matteis. The 3xHA-M1 human muscarinic receptor was a kind gift from Dr. Jurgen Wess (NIDDK, NIH). EGFP-GOLPH3 was amplified by PCR amplification from a human cDNA clone (SC112810, NM_022130.3) purchased from ORIGENE using primers containing XhoI and EcoRI restriction sites to insert into the pEGFP-C1 plasmid. All primers used in this study are listed in Table EV1.

## Cell culture

HEK293 cells stably expressing the AT1a rat AngII receptor (HEK293-AT1 (Hunyady et al, 2002)), PITPNA K/O, PITPNB K/O and Nir2 K/O cells described in (Kim et al, 2022) were maintained in Dulbecco's modified Eagle's medium (DMEM—high glucose, sodium pyruvate) containing 10% FBS and 1% penicillin-streptomycin. The cell line has been treated with Plasmocin prophylactic (InvivoGen, San Diego, CA) at 25 µg/ml for 1 week after thawing. The subsequent passages were maintained at 5 µg/ml of Plasmocin.

## Live-cell image acquisition, -processing, and analysis

About 350,000 cells were seeded on 30 mm glass bottom culture dishes (#1.5, Cellvis, Mountain View, CA) pre-coated with 0.01% poly-L-lysine solution (Sigma-Aldrich). Cells were transfected the next day with the indicated plasmid DNAs (0.1–0.2 µg/well) using Lipofectamine 2000 (2–5 µL/well; Invitrogen) using the manufacturer's protocol. After 1 day of transfection, the media was replaced with 1 ml modified Krebs–Ringer buffer (containing 120 mM NaCl, 4.7 mM KCl, 2 mM $CaCl_2$, 0.7 mM $MgSO_4$, 10 mM glucose, and 10 mM Na-Hepes, adjusted to pH 7.4) and cells were observed at room temperature with a Zeiss confocal microscope (LSM510, LSM710 or LSM 880 Airyscan) using a 63x Plan-Apochromat oil-immersion objective (N.A: 1.4). To quantify Golgi-localized EGFP-tagged lipid probes in HEK293 cells, z-stacks of fields with multiple cells were acquired in multi-position mode (3–4 positions) as a

time series from each separate dish. Maximum intensity projection, generated by the Zeiss Zen software, were then analyzed in Fiji software. Regions of interest (ROI) were selected after global thresholding, and the multi-measure function of the time series was used where the average pixel intensities were multiplied by the number of pixels covering the Golgi area, and for each individual cell, the initial values were taken as 100 percent. A typical dish yielded values obtained from about 30–40 cells that were averaged, and these values were then used to calculate the grand averages with S.E.M. from multiple dishes obtained in at least two independent experiments.

## BRET measurement

HEK293-AT1 cells (40,000/well) were seeded in a 200 µL total volume to white-bottom 96 well plates pre-coated with 0.01% poly-L-lysine solution (Sigma) and cultured overnight. Cells were then transfected with 0.1–0.25 µg of the specified BRET biosensor using Lipofectamine 2000 (0.5–1 µL/well) within OPTI-MEM (40 µL) according to the manufacturer's protocol. After 25–27 h of transfection, the cells were quickly washed before being incubated for 30 min in 50 µl of modified Krebs–Ringer buffer at 37 °C in ambient air. After the preincubation period, the cell-permeable luciferase substrate, coelenterazine h (40 µl, final concentration 5 µM), was added, and the signal from the mVenus fluorescence and sLuc luminescence were recorded using 485 and 530 nm emission filters over a 4 min baseline BRET measurement (15 s/cycle). Following the baseline recordings, where indicated, the plates were quickly unloaded for the addition of various treatments, which were prepared in a 10 µl volume of the modified Krebs–Ringer solution and added manually using a repeater pipet to achieve good mixing. Measurements were then continued for the indicated times. All measurements were performed in triplicate wells. BRET ratios (mVenus/Luciferase) were calculated for each well by dividing the 530-nm with the 485-nm intensity values, which, when indicated, were normalized to the baseline measurement. To facilitate the pooling of data from individual wells and between replicate experiments, the raw BRET ratios were processed using a simple moving average with a four-cycle interval across the BRET kinetic. The processed BRET ratios obtained from drug-treated wells were then normalized to internal vehicle controls.

## Lipidomics analyses

HEK293-AT1 cells (600,000 cells/well, passage 6–9) were plated onto 60 mm culture dishes and cultured for 2 days. VT01454 were added for 90 min in DMEM with high glucose medium with 10% serum and then cells were detached with trypsin-EDTA and centrifuged. The cell pellet was resuspended in PBS (at 6000 cells/µl) and frozen on dry ice for shipment to Lipotype GmBH (Drezden, Germany) for mass spectrometry-based lipid analysis as described in (Surma et al, 2021). Briefly, lipids were extracted using a chloroform/methanol procedure (Ejsing et al, 2009). Samples were spiked with internal lipid standard mixture containing: cholesterol ester 16:0 D7 (CE), diacylglycerol 17:0/17:0 (DAG), phosphatidate 17:0/17:0 (PA), phosphatidylcholine 15:0/18:1 D7 (PC), phosphatidylethanolamine 17:0/17:0 (PE), phosphatidylglycerol 17:0/17:0 (PG), phosphatidylinositol 16:0/16:0 (PI), phosphatidylserine 17:0/17:0 (PS), sphingomyelin 18:1;2/12:0;0 (SM) and triacylglycerol 17:0/

17:0/17:0 (TAG). After extraction, the organic phase was transferred to an infusion plate and dried in a speed vacuum concentrator. The dry extract was resuspended in 7.5 mM ammonium formate in chloroform/methanol/propanol (1:2:4; V:V:V). All liquid handling steps were performed using the Hamilton Robotics STARlet robotic platform with the Anti Droplet Control feature for organic solvent pipetting.

### MS data acquisition

Samples were analyzed by direct infusion on a QExactive mass spectrometer (Thermo Scientific) equipped with a TriVersa NanoMate ion source (Advion Biosciences). Samples were analyzed in both positive and negative ion modes with a resolution of Rm/z = 200 = 280,000 for MS and Rm/z = 200 = 17,500 for MSMS experiments, in a single acquisition. MSMS was triggered by an inclusion list encompassing corresponding MS mass ranges scanned in 1 Da increments (Surma et al, 2021). Both MS and MSMS data were combined to monitor CE, DAG, and TAG ions as ammonium adducts; PC as formate adducts; and PA, PE, PG, PI, and PS as deprotonated anions. MS was only used to monitor SM as formate adducts.

### Data analysis and post-processing

Data were analyzed with in-house developed lipid identification software based on LipidXplorer (Herzog et al, 2012; Herzog et al, 2011). Data post-processing and normalization were performed using an in-house developed data management system. Only lipid identifications with a signal-to-noise ratio >5, and a signal intensity fivefold higher than in corresponding blank samples were considered for further data analysis.

## Analysis of myo-[³H]inositol or [¹⁴C]-serine- labeled lipids

HEK293-AT1 cells (500,000 cells/dish) were plated on 12-well plates and cultured for 2 days. For myo-[³H]inositol labeling, the radioactive tracer (10 µCi/ml) was added for 1 h in inositol-free DMEM supplemented with 50 µM unlabeled myo-inositol with or without AngII (100 nM) and with our without VT01454 (100 nM). The steady-state level of inositol lipids were measured with 2 µCi/ml myo-[³H]inositol in 24 h labeling in inositol-free DMEM supplemented with 2% dialyzed serum and 50 µM unlabeled myo-inositol and VT01454 (100 nM) was treated for the last 30 min during labeling. For [¹⁴C]-serine labeling, cells were incubated with the radioactive tracer (0.5 µCi/ml) for 2 h in serine-free DMEM supplemented with or without VT01454. All reactions were terminated by the addition of ice-cold perchloric acid (to a final concentration of 5%), and cells were kept on ice for 10 min. After scraping, cells were centrifuged, and the pellet was processed to extract lipids by an acidic chloroform/methanol extraction (Nakanishi et al, 1995). The resulting lower organic phase was dried by nitrogen gas and followed by thin layer chromatography (TLC) using each solvent system either with chloroform/ methanol/ ammonia/water 70:70:4:16 (v/v), for inositol labeled lipids or chloroform/ methanol/ acetic acid/ water 85:38:5:8 (v/v) for serine-labeled lipids. For inositol lipids separation, TLC plates (silica gel 60 W, 20 × 20 cm, EMD Millipore) were pre-impregnated with an impregnating solution (75 mM Na-oxalate, 2 mM EDTA, 0.5% boric acid) and completely dried at 80'C in an oven. TLC plates with [¹⁴C]-labeled samples were exposed to X-ray films with multiple exposures while [³H]-labeled TLC plates were immersed in the solution (diphenyl-oxazole 5% in diethylether) and air dried to enhance the [³H]-signal before radiography.

## Protein expression, purification, and crystallization

The protein was expressed and purified as described in Tilley et al, 2004. For functional assays, wild-type PITPNA and its mutants (T58E, C94A, C94T) were expressed from the pET-19b plasmids. For structural analysis, the gene encoding full-length PITPNA was cloned into a modified pHIS2 vector containing an N-terminal His6x-tag followed by a SUMO tag. E. coli BL21 Star cells were transformed by these expression plasmids, and the cells were grown in 5052-ZYP autoinduction media. The cells were harvested and lysed by sonication in lysis buffer ((50 mM Tris pH 8, 300 mM NaCl, 20 mM imidazole, and 10% glycerol). Upon clearing the lysate by centrifugation, the protein was isolated by affinity chromatography on Ni-NTA resin and eluted in lysis buffer supplemented with 300 mM imidazole. The proteins for functional assays were further subjected to size exclusion chromatography (20 mM Tris 7.4, 300 mM NaCl, 10% glycerol, 3 mM β-mercaptoethanol), concentrated at 3–12 mg/ml and stored at −80 °C until needed.

His6x-SUMO tag was cleaved off from the protein for crystallization trials by the Ulp1 protease (4 °C overnight), the protein was bound on HiTrap Q HP column (Cytiva) in 20 mM Tris pH 8.0 and eluted with a gradient of 0–500 mM NaCl. The protein was further purified by size exclusion chromatography (SEC) on HiLoad 16/600 Superdex75 pg column (Cytiva) in SEC buffer (20 mM Tris pH 7.4). The protein was concentrated to 12 mg/ml in the size exclusion buffer (20 mM Tris pH 7.4), mixed with VT01454 at a 1:1.5 molar ratio (protein:inhibitor), and the mixture was incubated at room temperature for ~60 min. Crystallization was performed using the sitting drop vapor diffusion technique. Drops were created by mixing 150 nl of protein solution and 150 nl of reservoir solution of several commercial crystallization screens using a Mosquito robot (SPT Labtech). Crystals were obtained in 5 days in a condition with well solution of 100 mM imidazole pH 8, 10% PEG 8000. The crystals were cryo-protected in well solution supplemented with 20% glycerol and flash-frozen in liquid nitrogen. The crystallographic dataset was collected from a single crystal at the home source. The crystals diffracted to 2.3 Å and belonged to the orthorhombic P 2 $2_1$ $2_1$ space group. The data were integrated and scaled using XDS (Kabsch, 2010). The structure was solved by molecular replacement using the structure of PITP in complex with PI (PDB ID: 1T27) as the search model and further refined in Phenix (Afonine et al, 2012) and Coot (Emsley et al, 2010) to good geometry and R-factors (Table 1). Figures were generated with the PyMOL Molecular Graphics (Schrödinger, LLC). The atomic coordinates and structural factors were deposited in the Protein Data Bank (https://www.rcsb.org) under the PDB accession code 8PQO.

## In vitro PITP lipid binding assay

HEK293-AT1 cells (5,000,000 cells/dish) were cultured for 2 days in 100 mm culture dishes, and cells were labeled with 1.5 µCi/ml [$^{14}$C]acetic acid in 7 ml DMEM containing 2% dialyzed FBS for last 1 day. Cells were scraped and homogenized in 2 ml ice-cold buffer (250 mM sucrose, 10 mM Tris, pH 7.5, 1 mM EDTA, 1 mM dithiothreitol) with ten times trituration up-and-down through a 25 G syringe needle. Cell homogenates were then centrifuged briefly at 600 × g for 5 min to remove nuclei and cell debris. The postnuclear fraction was collected by centrifugation at 100,000 × g for 1 h to obtain the microsome pellet. Microsomes were resuspended in 300 µl of assay buffer (10 mM Hepes-Na, 50 mM NaCl, 1 mM

EDTA, pH 7.4, protease cocktail, 0.1 mM dithiothreitol). The recombinant PITPNA proteins (50 µg) and microsome (300 µg proteins) were mixed in 1 ml assay buffer and tumbled in a 1.5 ml Eppendorf tube for 30 min at room temperature. The reaction mixture was centrifuged at 200,000 × g for 1 h, and supernatant (PITPNA protein with bound lipids) and pellet (unbound membrane fractions) were processed to extract lipids by an acidic chloroform/methanol extraction as described previously (Nakanishi et al, 1995). The lower phase was dried and developed by TLC (chloroform/methanol/acetic acid/water 85:15:10:3, (v/v)). The same amounts of individual proteins were loaded on SDS-PAGE gel and stained with Coomassie Blue.

## Data availability

The atomic coordinates and structural factors were deposited in the Protein Data Bank (https://www.rcsb.org) under the PDB accession code 8PQO.

## Peer review information

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

## Acknowledgements

We would like to thank Dr. Radim Nencka (Institute of Organic Chemistry and Biochemistry AS CR, Prague 6, Czech Republic) and Dr. Matthew Shair (Department of Chemistry and Chemical Biology, Harvard University) for the MI-14 and OSW1 inhibitors, respectively. We are also thankful to Dr. Peter Varnai (Semmelweis University, Medical School, Budapest, Hungary), Dr.

Antonietta (Antonella) De Matteis (Telethon Institute of Genetics and Medicine, Naples, Italy), and Dr. Jurgen Wess (NIDDK, NIH, Bethesda, MD) for sharing DNA constructs. We would also like to express our thanks to Dr. Shamshad Cockcroft for the PITP antibodies. This work was partly funded by the Intramural Research Program of the *Eunice Kennedy Shriver* National Institute of Child Health and Human Development of the National Institutes of Health, Bethesda, MD, USA. (HHS|NIH|NICHD - Z01:HD000196-25). Confocal imaging was performed at the Microscopy & Imaging Core of the National Institute of Child Health and Human Development, NIH, with the kind assistance of Drs. Vincent Schram and Ling Yi. The research of A.E. and E.B. was funded by the project of the National Institute Virology and Bacteriology (Program EXCELES, Project No. LX22NPO5103) —Funded by the European Union—Next Generation Program. The Academy of Sciences of the Czech Republic RVO: 61388963 for its support of the Boura group is also acknowledged.

## Author contributions

**Yeun Ju Kim**: Conceptualization; Data curation; Formal analysis; Investigation; Methodology; Writing—review and editing. **Joshua G Pemberton**: Resources; Formal analysis; Validation; Investigation; Methodology; Writing—review and editing. **Andrea Eisenreichova**: Formal analysis; Investigation; Methodology. **Amrita Mandal**: Investigation; Methodology. **Alena Koukalova**: Investigation; Methodology. **Pooja Rohilla**: Investigation; Methodology. **Mira Sohn**: Resources; Data curation. **Andrei W Konradi**: Resources. **Tracy T Tang**: Resources; Writing—review and editing. **Evzen Boura**: Formal analysis; Funding acquisition; Writing—review and editing. **Tamas Balla**: Conceptualization; Data curation; Supervision; Funding acquisition; Visualization; Writing—original draft; Project administration.

## Disclosure and competing interests statement

Drs. Tracy T Tang and Andrei W Konradi are employees and shareholders of Vivace Therapeutics, Inc. The remaining authors declare no competing interests.

# Expanded View Figures

**Figure EV1.  Ability of recombinant PITPNA proteins to extract PI and PC from prelabeled membranes.**

(**A, B**) Membranes prepared from HEK293-AT1 cells prelabeled with [$^{14}$C]acetate overnight were incubated with purified recombinant PITPNA wild-type or mutated in the indicated residues. After centrifugation, to pellet the membranes, the supernatant was subjected to lipid extraction and TLC analysis as detailed in the Methods. Autoradiography films with different exposure times are shown. (**B**) PI and PC spots were quantified from two independent experiments using a Phosphor-Imager and normalized to the values obtained in the wild-type proteins. (**C**) The same amounts of PITPs used in the above lipid binding assays were also run on SDS gels and visualized with Coomassie staining. (**D, E**) Western Blot analysis of the various mutant mRFP-tagged PITPNA and PITPNB proteins expressed in the rescue experiments. Membranes were probed with specific antibodies against PITPNA and PITPNB, which were generous gift from Dr. Shamshad Cockcroft, as indicated.

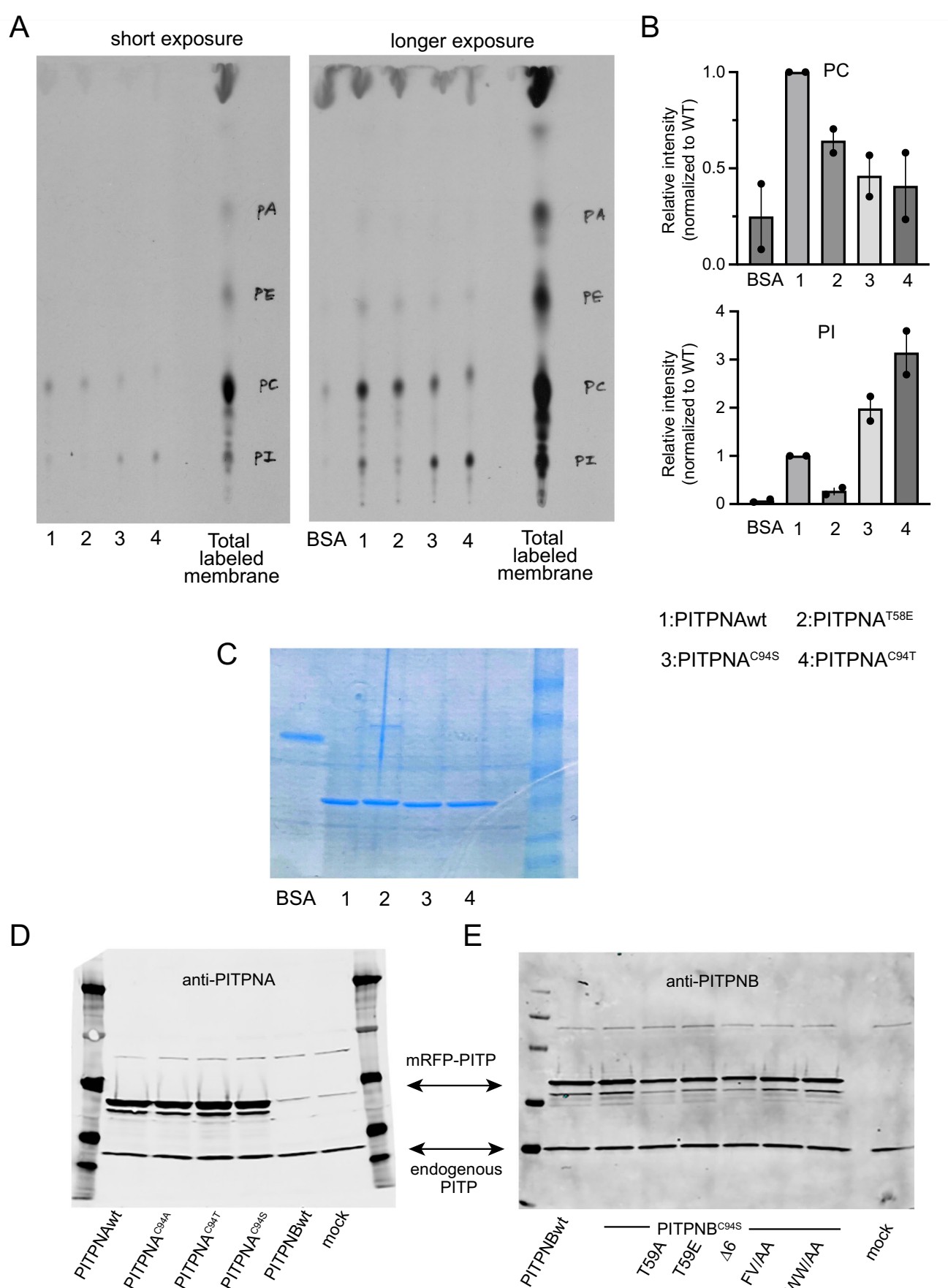

A

short exposure    longer exposure

PA
PE
PC
PI

1  2  3  4  Total labeled membrane    BSA  1  2  3  4  Total labeled membrane

B

PC

PI

1:PITPNAwt    2:PITPNA$^{T58E}$
3:PITPNA$^{C94S}$    4:PITPNA$^{C94T}$

C

BSA  1  2  3  4

D    anti-PITPNA

mRFP-PITP

endogenous PITP

PITPNAwt  PITPNA$^{C94A}$  PITPNA$^{C94T}$  PITPNA$^{C94S}$  PITPNBwt  mock

E    anti-PITPNB

PITPNBwt   PITPNB$^{C94S}$
T59A  T59E  Δ6  FV/AA  WW/AA   mock

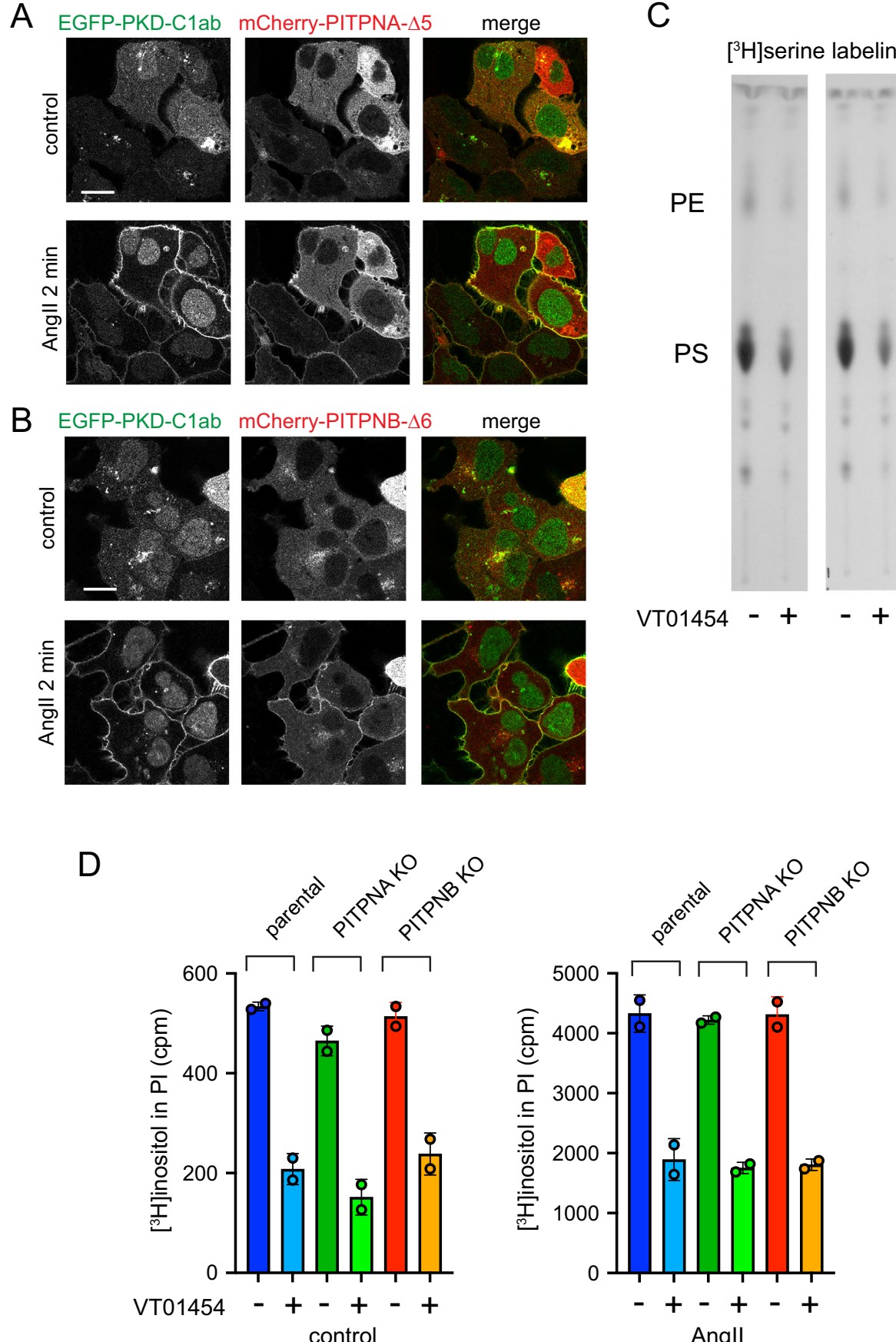

**Figure EV2. Co-localization of C-terminally truncated PITPs and a DAG sensor in HEK293-AT1 cells and effects of VT01454 on synthetic rates of PS and PI production.**

(A, B) Representative confocal images showing HEK293-AT1 cells transfected with the indicated constructs before (top rows) and after (bottom rows) stimulation with AngII (100 nM). Note the rapid increase in PM association of both the DAG sensor (PKD-C1ab) and the truncated PITPs. Scale bar 10 µm (note that all images in panels (A) and (B), respectively, show the same cells in the various channels before and after stimulation). (C) Incorporation of [$^{14}$C]-serine into cellular lipids in a 2 h incubation period shown as two biological replicates (see Methods for details). Note the inhibition of PS synthesis by treatment with VT01454 (100 nM), which was present throughout the labeling period. (D) Effects of VT01454 on the rate of PI labeling with myo-[$^{3}$H]inositol in a 1 h incorporation period either in the presence (right) or absence (left) of AngII. The results of two independent experiments are plotted. Parental HEK293-AT1 (blue) or their PITPNA (green) or PITPNB (red/orange) K/O derivatives are shown. Note the different scales of the two graphs.

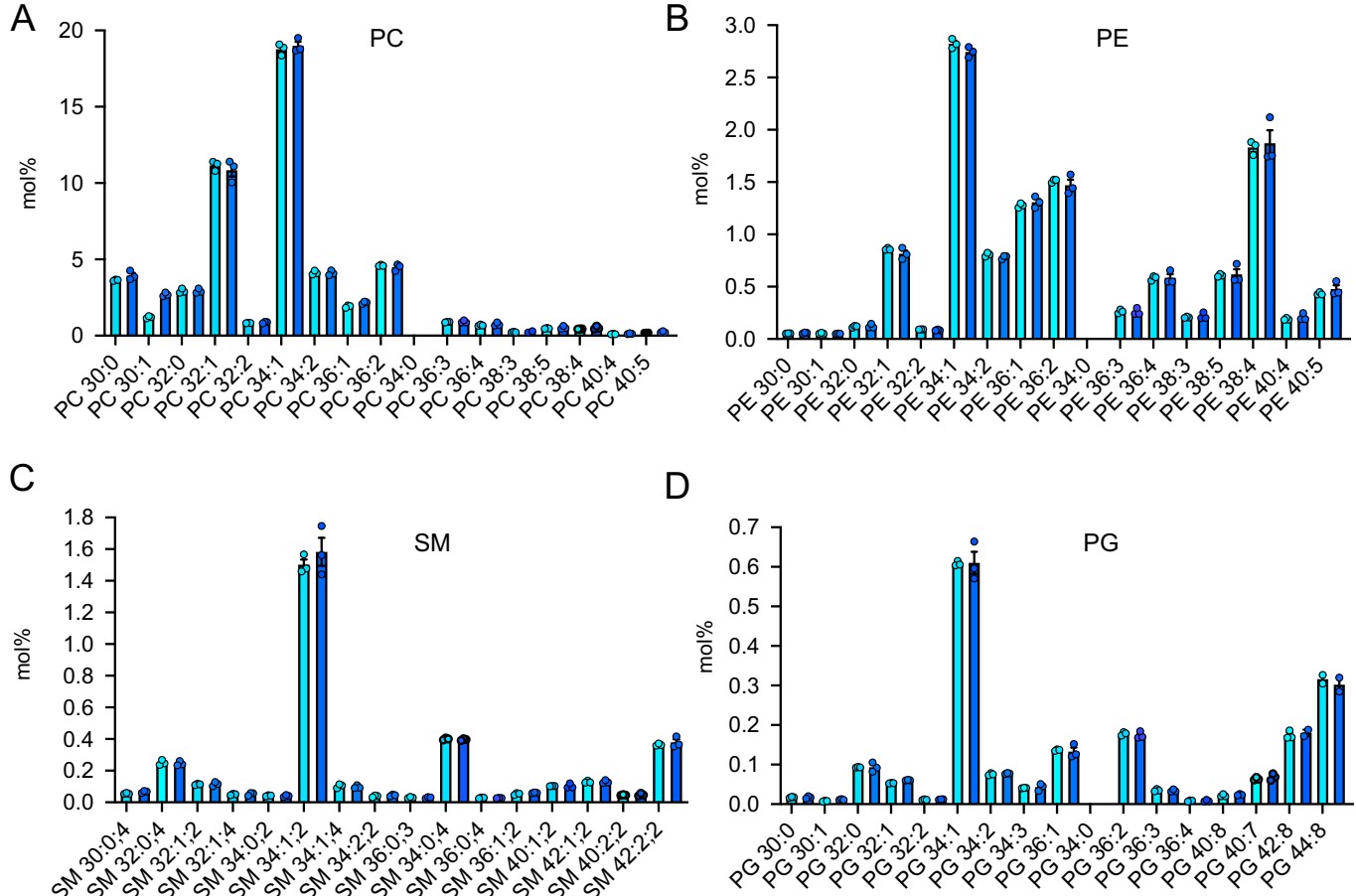

**Figure EV3. Lipidomics analyses of HEK293-AT1 cells treated with VT01454.**

HEK293-AT1 cells were treated with VT01454 (100 nM) or DMSO for 90 min and prepared for lipidomic analysis, which were performed by Lipotype as described in the Methods. Means ± SEM and the individual data points are shown from biological triplicates from one experiment that was repeated with essentially the same results. Dark columns show VT01454-treated cells. No significant changes were observed in either of the four lipid classes shown in panels (**A–D**).

