## [Peer Review File · The EMBO Journal]

NON-VESICULAR PHOSPHATIDYLINOSITOL TRANSFER PLAYS CRITICAL ROLES IN DEFINING ORGANELLE LIPID COMPOSITION

Yeun Ju Kim, Joshua Pemberton, Andrea Eisenreichova, Amrita Mandal, Alena Koukalova, Pooja Rohilla, Mira Sohn, Andrei Konradi, Tracy Tang, Evzen Boura, and Tamas Balla

Corresponding author(s): Tamas Balla (ballat@mail.nih.gov)

Review Timeline:

Submission Date:	11th Sep 23
Editorial Decision:	20th Oct 23
Revision Received:	6th Feb 24
Editorial Decision:	28th Feb 24
Revision Received:	12th Mar 24
Accepted:	21st Mar 24

Editor: William Teale

Transaction Report:

Dear Tamas,

Thank you again for the submission of your manuscript entitled "Non-vesicular phosphatidylinositol transfer plays critical roles in defining organelle lipid composition" (EMBOJ-2023-15582) and for your patience during the review process. We have now received reports from three referees, which I copy below.

As you can see from their comments, while referee #1 points to the possibility of off-target effects and requests a series of tighter controls, referees #2 and #3 both state that your data are important and well-founded.

Based on the overall interest expressed in the reports, I would like to invite you to address the comments of all referees in a revised version of the manuscript. I should add that it is The EMBO Journal policy to allow only a single major round of revision and that it is therefore important to resolve the main concerns at this stage. I believe the concerns of the referees are reasonable and addressable, but please contact me, especially if you would like to have a Zoom call about the extra bench-work that is requested. I would also be happy to talk over any other questions you may have, need further input on the referee comments or if you anticipate any problems in addressing any of the points listed. Please, follow the instructions below when preparing your manuscript for resubmission.

I would also like to point out that as a matter of policy, competing manuscripts published during this period will not be taken into consideration in our assessment of the novelty presented by your study ("scooping" protection). We have extended this 'scooping protection policy' beyond the usual 3 month revision timeline to cover the period required for a full revision to address the essential experimental issues. Please contact me if you see a paper with related content published elsewhere to discuss the appropriate course of action.

Again, please contact me at any time during revision if you need any help or have further questions.

Thank you very much again for the opportunity to consider your work for publication. I look forward to your revision.

Best regards,

William

William Teale, Ph.D.
Editor
The EMBO Journal

When submitting your revised manuscript, please carefully review the instructions below and include the following items:

- 1) a .docx formatted version of the manuscript text (including legends for main figures, EV figures and tables). Please make sure that the changes are highlighted to be clearly visible.
- 2) individual production quality figure files as .eps, .tif, .jpg (one file per figure).
- 3) a .docx formatted letter INCLUDING the reviewers' reports and your detailed point-by-point response to their comments. As part of the EMBO Press transparent editorial process, the point-by-point response is part of the Review Process File (RPF), which will be published alongside your paper.
- 4) a complete author checklist, which you can download from our author guidelines ([https://wol-prod-cdn.literatumonline.com/pb-assets/embo-site/Author Checklist%20-%20EMBO%20J-1561436015657.xlsx](https://wol-prod-cdn.literatumonline.com/pb-assets/embo-site/Author%20Checklist%20-%20EMBO%20J-1561436015657.xlsx)). Please insert information in the checklist that is also reflected in the manuscript. The completed author checklist will also be part of the RPF.
- 5) Please note that all corresponding authors are required to supply an ORCID ID for their name upon submission of a revised manuscript.
- 6) We require a 'Data Availability' section after the Materials and Methods. Before submitting your revision, primary datasets produced in this study need to be deposited in an appropriate public database, and the accession numbers and database listed under 'Data Availability'. Please remember to provide a reviewer password if the datasets are not yet public (see

<https://www.embopress.org/page/journal/14602075/authorguide#datadeposition>). If no data deposition in external databases is needed for this paper, please then state in this section: This study includes no data deposited in external repositories. Note that the Data Availability Section is restricted to new primary data that are part of this study.

Note - All links should resolve to a page where the data can be accessed.

8) For data quantification: please specify the name of the statistical test used to generate error bars and P values, the number (n) of independent experiments (specify technical or biological replicates) underlying each data point and the test used to calculate p-values in each figure legend. The figure legends should contain a basic description of n, P and the test applied. Graphs must include a description of the bars and the error bars (s.d., s.e.m.).

9) We would also encourage you to include the source data for figure panels that show essential data. Numerical data can be provided as individual .xls or .csv files (including a tab describing the data). For 'blots' or microscopy, uncropped images should be submitted (using a zip archive or a single pdf per main figure if multiple images need to be supplied for one panel). Additional information on source data and instruction on how to label the files are available at .

10) We replaced Supplementary Information with Expanded View (EV) Figures and Tables that are collapsible/expandable online (see examples in <https://www.embopress.org/doi/10.15252/embj.201695874>). A maximum of 5 EV Figures can be typeset. EV Figures should be cited as 'Figure EV1, Figure EV2" etc. in the text and their respective legends should be included in the main text after the legends of regular figures.

12) Our journal encourages inclusion of *data citations in the reference list* to directly cite datasets that were re-used and obtained from public databases. Data citations in the article text are distinct from normal bibliographical citations and should directly link to the database records from which the data can be accessed. In the main text, data citations are formatted as follows: "Data ref: Smith et al, 2001" or "Data ref: NCBI Sequence Read Archive PRJNA342805, 2017". In the Reference list, data citations must be labeled with "[DATASET]". A data reference must provide the database name, accession number/identifiers and a resolvable link to the landing page from which the data can be accessed at the end of the reference. Further instructions are available at .

We realize that it is difficult to revise to a specific deadline. In the interest of protecting the conceptual advance provided by the work, we recommend a revision within 3 months (18th Jan 2024). Please discuss the revision progress ahead of this time with the editor if you require more time to complete the revisions. Use the link below to submit your revision:

Referee #1:

PI transfer proteins (PITPs) are thought to transfer PI from the ER, where it is synthesized, to downstream compartments, where it is converted into phosphoinositides, in particular PI4P, which can be used to power transport of other lipids. They also transport PC and may mediate PI/PC exchange. Class I PITPs, PITPNA & PITPNB are soluble single-domain proteins; several structures of PITPNA are available. They can bind both PI and PC and may mediate PI/PC exchange; however, their precise function is a matter of debate and different models exist, including that they don't transfer lipid between different membranes. Class II PITPs are multi-domain proteins and can directly target contact sites. Kim et al. take the advantage of a recently identified PITPN inhibitor VT01454 (Li et al., 2022) to study the function of the class I PITPs. They determine the structure of PITPNA in complex with the inhibitor, which is in agreement with molecular modeling performed by Li et al. Then they use their well-established BRET assay, different lipid probes in combination with other inhibitors, and lipidomic analyses to assess the effect of PITP inhibition on cellular lipid levels and distribution. They conclude that PITP inhibition has wide-spread effects, some of which are in agreement with existing models of PITP function, others are more surprising.

The aim of this work was to use acute inhibition by VT01454 in order to directly assess the function of class I PITPs inside the cell. The experiments presented in the manuscript are of good quality and for the most part convincing, taking advantage of established assay in the Balla group and their great expertise in this area, although some important controls are missing. The observed effects in cells are clear and significant, and some are very interesting; however, the results are largely descriptive and do not lead to a more precise mechanistic understanding of PITP function. Despite the relatively short time of inhibition, many of the observed effects are likely indirect, which is an inherent (and well-known) problem of studying the highly interconnected lipid transport/metabolism pathways.

The manuscript would also greatly benefit by some simplification and reorganization of the figures + inclusion of obvious (even if un-interesting) controls. The authors use a lot of different pharmacological tools and PITP mutants that relate to observations from many different studies; it is quite difficult to keep up with them. The figures could be more self-explanatory (Figure 3 is presented as an intelligence test, and following Figure 5 requires constant switching between the text of the Results section and the figure). A more complete model figure summarizing all effects would be very helpful.

Major comments:

1. Given the diverse effects on lipid distribution observed after VT01454 treatment, some of which are unexpected, are the authors sure that VT01454 does not affect the activity of any other proteins besides PITPNA/B? In fig. 2A, the levels of PI4P and PI4,5P2 are about equally affected in WT, PITPNA KO and PITPNB KO cells. It would be important to show that PITPNA/B double KO are not affected by VT treatment.

The authors could also perform auxin-mediated depletion of PITPNA/B and compare the effects with VT01454 treatment, or at

least provide some direct comparison between the effect of VT treatment and PITP down-regulation.

2. It would be informative to have a better understanding of the effect of VT01454 on PITP function in vitro or in cells. Is the inhibition irreversible or can the inhibitor be washed out? Does VT equally affect the affinity of PITP for PC and PI, or is the binding of one lipid more affected than the other? Does VT treatment affect the localization of PITP A/B in the cells? (this is partially shown in Fig 6 for PITPB, but only after VT treatment; why are controls without VT treatment (+/- AngII) not shown?)
3. In the overexpression rescue experiments in Fig.4, are the different mutants expressed at the same level?
4. The authors suggest that VT blocks PITPs in an open conformation, and then use delta-N PITP mutants that are also supposed to represent the open conformation and show increased binding to DAG. It would be useful to have a direct comparison between VT treatment in delta-N mutants (combining figures 5 and 6 gives some comparison, but the conditions are not the same. Also, showing colocalization with a DAG probe would be helpful.
5. The authors show that VT treatment affects PI4P levels at the PM but not at the Golgi. This was already shown by Li et al., so in this respect, it is not surprising. But it is surprising given that PITPs are generally thought to function between the ER and the Golgi, and they are reported to localize to the cytosol or to the Golgi. What is the explanation for the pronounced effects at the PM?
6. Lipid analyses:
 - No information is provided on how the lipidomic analyses were performed.
 - Is there any difference in cholesterol levels between +/- inhibitor?
 - The effect on neutral lipids is very pronounced; it's not very clear why this data is shown in supplementary information.
6. Did the authors test shorter times of VT treatment?

Referee #2:

This study addresses a long-standing mystery about how PI transfer proteins (PITPs) function in cells. The proteins facilitate PI exchange between membrane in vitro, but, as the authors explain, "proof that PITPs are indeed responsible for delivering PI to various membranes [in cells] has been lacking partially due to functional redundancies between the various PITPs that are found in higher organisms." Knock down and knock out studies of cells lacking PITPs have substantially blocked PI transport in cells. This could be because compensatory occur. One way around this problem is to rapidly inhibit or deplete PITPs, which is what this study does. It uses the Class I PITPs inhibitor VT01454. The structure of the PITPNA with VT01454 is solved, which reveals how it blocks binding to PI and PC, which is also bound by PITPNA. It goes on to characterize the how rapid inhibition of PITPNA affects phosphoinositide metabolism globally and at the plasma membrane (PM) and Golgi, where PI4P and PI(4,5)P2 are primarily synthesized. The results make a compelling case the Class I PITPs affect the formation of PI4P and perhaps other monophosphorylated PI. This most clearly seen at the PM but interestingly not at the Golgi. Surprisingly, PI(4,5)P2 formation is not affected even though it is produced from PI4P produced at the PM. These are a fascinating and important set of findings. The work is well done and convincing. There are, however, some concerns that should be addressed before publication.

1. The study says that inhibition of PITPs alters the abundance of lipids in addition to PI and phosphoinositides. The evidence is not terribly convincing. The changes in non-PI lipids are quite modest and it is not clear how directly related they are to changes in PI metabolism. Many of the changes are in neutral lipids (cholesteryl ester and triglyceride), which often happen when cells are stressed or growth slows. Whether VT01454 affects grow, metabolic rate, or induces ER stress has not been determined. Also, the lipid analysis was performed with cells treated with VT01454 for 90 minutes whereas the cells used for the PI experiments were treated for only 30 minutes.
2. The experiments on membrane association of PITPs (Figs. 5 and 6) are well done, but it is not clear what they add to the story. This should be clarified, or the data removed.
3. The ability of recombinant, mutant PITPNA to bind lipids needs more controls. The results in Fig. S1 should be quantified. Also, expression levels of the mutant PITPs in cells should be determined.

Referee #3:

Over the last decades, researchers have been identifying lipid transfer proteins with important roles in membrane homeostasis and lipid signaling, many with overlapping cargo specificities and even localizations; very clear is that phosphoinositide metabolism and lipid metabolism more generally is extremely complex and that deciphering its workings will require careful studies--such as the ones in this paper. The authors present a first comprehensive analysis of the role of Class I PITPs in

cellular lipid metabolism using pharmacological approaches. They characterize and leverage a new class I PITP inhibitor VT01454 in showing that class I PITPs are key in maintaining PI4P levels at the PM and also at Rab7-positive (late endosomal) compartments. They show that PITPs do not play a major role in regulating PI levels at the Golgi or early (Rab5-positive) endosomes, and do not play a major role in supporting PI(4,5)P₂ replenishment following strong PLC stimulation (except when class I PITPs are not present). The authors have also shown that class I mediated PI transfer is critical in regulating levels of multiple lipid classes other than PI, highlighting how very intricate lipid metabolism and homeostasis is...and how much studies of this type will be needed. The conclusions are significant and all well-founded.

The authors are recognized experts in the development and utilization of pharmacological approaches in understanding phosphoinositide signaling and homeostasis, and the manuscript as presently written is for other experts in such approaches. However, it is very difficult reading for non-experts. The authors should include a summary sentence at the end of each section explaining the section's major finding. Such summary statements are already present in some sections--but not all.

Other requests to make the manuscript accessible to non-experts or minor points:

--On page 7, in "Pharmacological blockade of Class I PITP's selectively affects cellular PIPs pools" Could the authors expand the explanation of "BRET" in Fig 2B to more than one sentence in the legend, explaining that the method is shown schematically?

-- On page 8, same section, it was not clear that PI4P levels following VT01454 treatment did decrease to "below initial levels" after the initial sharp decline; the decline seems barely significant. But that class I PITPs do contribute Rab7+ compartments was clear when the cells were treated with OSBP-inhibitor, so their conclusion that class I PITP's play a role is well justified.

--Right after, they state that there is only a minor effect on PI3P pools in Rab5+ compartments. The authors later speculate that PI levels in early endosomal compartments most likely rely heavily on plasma membrane levels--perhaps they might already provide some kind of interpretation here?

--This section as well as the section "Expression of drug resistant PITPs can rescue the lipid changes caused by VT01454" might benefit from summary sentences at the conclusion (in addition to the section titles).

--The reviewer is very confused as to the purpose of the section "Diacylglycerol promotes membrane association of the "open" conformation of class I PITP's". Many PITP constructs are investigated, but there already seems to be a lot known regarding behavior in vitro. Is the new information that the constructs exhibit expected behavior in vivo? Perhaps the section is more intelligible to readers knowledgeable on nitty-gritty details of PITP function; again, this section seems to be written for experts, in this case PITP, rather for an audience generally interested in membrane homeostasis more generally.

It is also not clear how this section fits in with the rest of the manuscript, except characterization at the very end that indicates that VT01454 binding is consistent with stabilizing an open conformation that binds membranes. It would be helpful if the authors could explain how studies on pages 9+10 other than those pertaining to VT01454 are relevant to the story they are telling (or are they just packing in a whole bunch of experiments that they do not know how else to publish?)...

Also, throughout the whole section, the reviewer was left wondering how the C-terminus of the PITPs so much affects membrane localization. Perhaps an explanation could be offered?

Referee #1:

PI transfer proteins (PITPs) are thought to transfer PI from the ER, where it is synthesized, to downstream compartments, where it is converted into phosphoinositides, in particular PI4P, which can be used to power transport of other lipids. They also transport PC and may mediate PI/PC exchange. Class I PITPs, PITPNA & PITPNB are soluble single-domain proteins; several structures of PITPNA are available. They can bind both PI and PC and may mediate PI/PC exchange; however, their precise function is a matter of debate and different models exist, including that they don't transfer lipid between different membranes. Class II PITPs are multi-domain proteins and can directly target contact sites. Kim et al. take the advantage of a recently identified PITPN inhibitor VT01454 (Li et al., 2022) to study the function of the class I PITPs. They determine the structure of PITPNA in complex with the inhibitor, which is in agreement with molecular modeling performed by Li et al. Then they use their well-established BRET assay, different lipid probes in combination with other inhibitors, and lipidomic analyses to assess the effect of PITP inhibition on cellular lipid levels and distribution. They conclude that PITP inhibition has wide-spread effects, some of which are in agreement with existing models of PITP function, others are more surprising.

Thank you for this summary. Here we would like to point out that our X-ray structure does not agree with the modeling data published by Li et al. They modeled the inhibitor in the closed PITP structure and missed most of the important hydrogen bonds. Our structure clearly shows that VT01454 binding can stabilize the open conformation, and we present the real structure with an accurate description of the molecular interactions, which is a major advance over the published model.

The aim of this work was to use acute inhibition by VT01454 in order to directly assess the function of class I PITPs inside the cell. The experiments presented in the manuscript are of good quality and for the most part convincing, taking advantage of established assay in the Balla group and their great expertise in this area, although some important controls are missing. The observed effects in cells are clear and significant, and some are very interesting; however, the results are largely descriptive and do not lead to a more precise mechanistic understanding of PITP function. Despite the relatively short time of inhibition, many of the observed effects are likely indirect, which is an inherent (and well-known) problem of studying the highly interconnected lipid transport/metabolism pathways.

The manuscript would also greatly benefit by some simplification and reorganization of the figures + inclusion of obvious (even if un-interesting) controls. The authors use a lot of different pharmacological tools and PITP mutants that relate to observations from many different studies; it is quite difficult to keep up with them. The figures could be more self-explanatory (Figure 3 is presented as an intelligence test, and following Figure 5 requires constant switching between the text of the Results section and the figure). A more complete model figure summarizing all effects would be very helpful.

We have made a faithful effort to simplify the Figures and describe the results in a more accessible way. This included a complete reorganization of the manuscript with a different order in which the results are presented. We have also removed several cartoons from the Figures to make them easier to follow. We expanded the Figure legends and added a short summary conclusion at the end of each sub-section in the Results. We believe that the cartoon describing the processes that contribute to PI transport between the PM and the ER shown in revised Fig. 7A, is a good representative of how this transport may shape membrane PI and PI4P levels. Therefore, we decided against adding another summary Figure.

Major comments:

1. Given the diverse effects on lipid distribution observed after VT01454 treatment, some of which are unexpected, are the authors sure that VT01454 does not affect the activity of any other proteins besides PIPNA/B? In fig. 2A, the levels of PI4P and PI4,5P2 are about equally affected in WT, PIPNA KO and PIPNB KO cells. It would be important to show that PIPNA/B double KO are not affected by VT treatment.

We do appreciate the Reviewer's positive comments. As with many inhibitors, one can never be sure that there are no off-target effects (which is also true for genetic manipulations). However, the fact that the inhibitor is potent at the sub-micromolar concentration, its effects are reversed by expressing a drug resistant mutant PIP, and that the highly homologous Class II PIP is not touched by the inhibitor, make us fairly confident that the reported effects are due to a specific inhibition of the Class I PIPs. In fact, we have attempted to make a mutant Class II PIP that would be sensitive to the VT01454 compound and despite the extreme conservation between their PIP domains and those of Class I PIPs, so far, we have not been able to make the Class II PIP protein VT01454-sensitive.

We also want to point out that in the paper of Li et al (2022), the authors have performed a pull-down assay using the inhibitor and while they have identified a few proteins that were detectable, PIPs were by far the most enriched proteins.

We also want to emphasize, that this inhibitor allows us, for the first time, to investigate the role of Class I PIPs in PI distribution, a question that several groups have attempted to address before, but because of the redundancy of the two proteins, and the fact that double knockout cells (or animals) are not viable, these efforts (including ours) have had very limited success. In fact, as we reported earlier (Kim et al. 2022, PMID: 35712788) knocking down PIPNA in PIPNB knockout cells, or doing the opposite, could not get us closer to analyze the contribution of Class I PIPs to PI4P or PIP₂ maintenance.

The authors could also perform auxin-mediated depletion of PIPNA/B and compare the effects with VT01454 treatment, or at least provide some direct comparison between the effect of VT treatment and PIP down-regulation.

We respectfully argue that the degron approach would also suffer from similar problems, including non-viability of double knockouts, and that the secondary effects that would be even more likely to pose a problem with a several hours protocol for the complete degradation of the PIPs rather than the several minutes of inhibitor treatment.

2. It would be informative to have a better understanding of the effect of VT01454 on PIP function in vitro or in cells. Is the inhibition irreversible or can the inhibitor be washed out? Does VT equally affect the affinity of PIP for PC and PI, or is the binding of one lipid more affected than the other? Does VT treatment affect the localization of PIP A/B in the cells? (this is partially shown in Fig 6 for PIPB, but only after VT treatment; why are controls without VT treatment (+/- AngII) not shown?)

We thank the Reviewer for these questions. The effect of VT01454 is not reversible by simple washout, as it covalently binds to the C95 residue on the bottom of the lipid binding pocket, which the inhibitor fully occupies (as shown in the X-ray structure). Because of its fully obstructing the lipid binding pocket, it prevents the binding of any ligands, be it PC, PI or any other lipid for that matter. We did show that VT01454-treatment causes localization of the PIPs to membranes. This membrane binding develops

slowly and is sped up by AngII treatment, presumably because the lipid exchange cycle is activated when PLC depletes PI4P and PIP₂ in the plasma membrane. We have stated that AngII treatment alone does not make either of the Class I PITPs associate with the membrane in a visible manner and we showed that a VT01454-resistant PITP also shows no membrane localization after treatment with the inhibitor even when stimulated with AngII. Lastly, we have now included images of the cells before the treatment with the inhibitor in new Fig. 6B-D. We tried to describe these points more explicitly in the revised manuscript.

3. In the overexpression rescue experiments in Fig.4, are the different mutants expressed at the same level?

This is an important point raised by the Reviewer. We have now included in the Extended Fig. EV1D,E the expression levels of the various “rescue” constructs as measured by Western Blotting, under the same conditions that they were used in the rescue experiments.

4. The authors suggest that VT blocks PITPs in an open conformation, and then use delta-N PITP mutants that are also supposed to represent the open conformation and show increased binding to DAG. It would be useful to have a direct comparison between VT treatment in delta-N mutants (combining figures 5 and 6 gives some comparison, but the conditions are not the same. Also, showing colocalization with a DAG probe would be helpful.

We now included the co-localization images of PITP Δ 5/6 and our DAG sensor in Extended Fig. EV2A,B. The membrane association response of the truncated PITP constructs is more robust after AngII stimulation than that of the wild-type protein after VT treatment followed by AngII stimulation. VT01454-treatment alone has only a small effect and even that develops slowly (over 30 min). In that respect the pictures shown in Figs 5 and 6 are quite informative. We tried to state this more clearly in the Discussion (page 21).

5. The authors show that VT treatment affects PI4P levels at the PM but not at the Golgi. This was already shown by Li et al., so in this respect, it is not surprising. But it is surprising given that PITPs are generally thought to function between the ER and the Golgi, and they are reported to localize to the cytosol or to the Golgi. What is the explanation for the pronounced effects at the PM?

It is true that Li et al. has reported a brief assessment of the effects of the inhibitor on the various membrane compartments. However, those were based on simple imaging using some of the lipid probes and did not follow kinetic changes using our more quantitative methods. The lack of VT01454 effect on Golgi PI4P, indeed, was surprising to us to the extent that we spent a considerable effort to analyze Golgi PI4P changes in response to PI4KB inhibition (even in the PI4K2A knockout background) to evaluate whether PI4P levels in the Golgi depend on sustained generation of PI4P from PI. Since that Golgi has almost the highest level of PI (as shown in our previous study, Pemberton et al. PMID: 32211894) we reason that it will take a longer time to drop Golgi PI4P levels to the point that it becomes a limiting factor for PI4P synthesis. As for plasma membrane (PM) PI4P, the recently identified ORP5/8-mediated PI4P-PS exchange cycle between the ER and the PM (Chung et al. PMID: 26206935, together with the low steady-state level of PI in the PM (again, shown in our previous study, Pemberton et al. PMID: 32211894), make PI4P generation in the PM highly dependent on PI delivery, which we show is performed by Class I PITPs.

6. Lipid analyses:

- No information is provided on how the lipidomic analyses were performed.

Our mass spectrometry based lipidomics was performed by Lipotype (Germany). We have now included the details provided by Lipotype in the Method section.

- Is there any difference in cholesterol levels between -/+ inhibitor?

Cholesterol measurement was not included in the mass spectrometry analysis. However, using two different cholesterol sensors, including a recently developed series for free cholesterol (Koh et al. PMID: 37880244), we did not see any obvious effect of the distribution of the sensor in response to treatment with the inhibitor.

- The effect on neutral lipids is very pronounced; it's not very clear why this data is shown in supplementary information.

We have now moved these panels into a main Figure (new Fig. 8E,F).

6. Did the authors test shorter times of VT treatment?

Most of our studies are time-lapse, so we assume the Reviewer was referring to the lipidomic analysis. We have not done shorter treatments. Our choice of exposure time was based on the expectation that it will take some time before a meaningful change in the whole cell lipidome would develop. Given the time and cost associated with these analyses, we did not do a more detailed time course.

Referee #2:

This study addresses a long-standing mystery about how PI transfer proteins (PITPs) function in cells. The proteins facilitate PI exchange between membrane in vitro, but, as the authors explain, "proof that PITPs are indeed responsible for delivering PI to various membranes [in cells] has been lacking partially due to functional redundancies between the various PITPs that are found in higher organisms." Knock down and knock out studies of cells lacking PITPs have substantially blocked PI transport in cells. This could be because compensatory occur. One way around this problem is to rapidly inhibit or deplete PITPs, which is what this study does. It uses the Class I PITPs inhibitor VT01454. The structure of the PITPNA with VT01454 is solved, which reveals how it blocks binding to PI and PC, which is also bound by PITPNA. It goes on to characterize the how rapid inhibition of PITPNA affects phosphoinositide metabolism globally and at the plasma membrane (PM) and Golgi, where PI4P and PI(4,5)P2 are primarily synthesized. The results make a compelling case the Class I PITPs affect the formation of PI4P and perhaps other monophosphorylated PI. This most clearly seen at the PM but interestingly not at the Golgi. Surprisingly, PI(4,5)P2 formation is not affected even though it is produced from PI4P produced at the PM. These are a fascinating and important set of findings. The work is well done and convincing. There are, however, some concerns that should be addressed before publication.

We appreciate the Reviewer's positive comments of our manuscript.

1. The study says that inhibition of PITPs alters the abundance of lipids in addition to PI and phosphoinositides. The evidence is not terribly convincing. The changes in non-PI lipids are quite modest

and it is not clear how directly related they are to changes in PI metabolism. Many of the changes are in neutral lipids (cholesteryl ester and triglyceride), which often happen when cells are stressed, or growth slows. Whether VT01454 affects growth, metabolic rate, or induces ER stress has not been determined. Also, the lipid analysis was performed with cells treated with VT01454 for 90 minutes whereas the cells used for the PI experiments were treated for only 30 minutes.

We have done experiments related to the Reviewer's comments.

1. We have tested using Western Blot analysis whether VT-treatment causes ER stress and found that none of the ER-stress markers shown here indicate that this would be the case (see below).

2. We also tested if there are lipid droplets that change in response to VT treatment and found no obvious effects.
3. We also treated cell overnight, which basically killed the cells, but cells expressing the VT01454-resistant PITPs rescued cells (PITPNA being somewhat more effective than PITPNB).

While some lipid changes are indeed small, some are more robust (e.g. : PA 32:0 and 34:1 and 34:2 increase almost two fold, while none of the saturated PA species show a change). Similarly, (DAG 30:0 and 32:0 shows a sizable decrease, while the rest remains unchanged). We agree with the Reviewer that some of these changes may be indirect, but we trust that reporting of them is quite important as these are the first data to analyze the effects of interfering with PITP function in intact cells on the whole cell lipidome. These data already directed our attention to pathways that have not been previously considered. As for the choice of time point selected for lipidomics analyses, see our reply under point 6 of Reviewer 1.

2. The experiments on membrane association of PITPs (Figs. 5 and 6) are well done, but it is not clear what they add to the story. This should be clarified, or the data removed.

We believe that the membrane association of PITPs is an important question that has been studied *in vitro*, yet is still puzzling. We tried to integrate this part of the study better within the revised manuscript with a clearer rationalization for these studies and changing the order in which the results are presented. As part of this effort, we have broken up the original description of some experiments into shorter parts with better articulated aims and conclusions.

3. The ability of recombinant, mutant PITPNA to bind lipids needs more controls. The results in Fig. S1 should be quantified. Also, expression levels of the mutant PITPs in cells should be determined.

We have now included quantification of the results of experiments shown in Extended Fig. EV1 and included the WB on the expression level of the various rescue constructs (now shown in Extended Fig. EV1D,E). Since we had both negative and positive controls in the recombinant protein experiments, which were intended only to assess the effect of the C94S mutation on PI and PC binding, we assume that the only control the Reviewer had in mind was to test the effect of VT01454 on PC or PI binding. Given the covalent nature of the inhibition and the structure showing the inhibitor occupying the lipid binding cavity, it is clear that the inhibitor blocks the binding of any lipids. The effect of the other mutations on PI and PC binding and transport of Class I PITPs has been extensively documented in the cited literature.

Referee #3:

Over the last decades, researchers have been identifying lipid transfer proteins with important roles in membrane homeostasis and lipid signaling, many with overlapping cargo specificities and even localizations; very clear is that phosphoinositide metabolism and lipid metabolism more generally is extremely complex and that deciphering its workings will require careful studies--such as the ones in this paper. The authors present a first comprehensive analysis of the role of Class I PITPs in cellular lipid metabolism using pharmacological approaches. They characterize and leverage a new class I PITP inhibitor VT01454 in showing that class I PITPs are key in maintaining PI4P levels at the PM and also at Rab7-positive (late endosomal) compartments. They show that PITPs do not play a major role in regulating PI levels at the Golgi or early (Rab5-positive) endosomes, and do not play a major role in supporting PI(4,5)P₂ replenishment following strong PLC stimulation (except when class I PITPs are not present). The authors have also shown that class I mediated PI transfer is critical in regulating levels of multiple lipid classes other than PI, highlighting how very intricate lipid metabolism and homeostasis is...and how much studies of this type will be needed. The conclusions are significant and all well-founded.

Thank you for the kind words and the positive comments on our work.

The authors are recognized experts in the development and utilization of pharmacological approaches in understanding phosphoinositide signaling and homeostasis, and the manuscript as presently written is for other experts in such approaches. However, it is very difficult reading for non-experts. The authors should include a summary sentence at the end of each section explaining the section's major finding. Such summary statements are already present in some sections--but not all.

We have included summary statements after each of the sections and tried to make the paper more accessible for readers less experienced in this field. As pointed out in our response to Reviewers 1 and

2, we have reorganized the sequence in which the Results are presented to make the various parts more logically linked.

Other requests to make the manuscript accessible to non-experts or minor points:

--On page 7, in "Pharmacological blockade of Class I PITP's selectively affects cellular PPIs pools" Could the authors expand the explanation of "BRET" in Fig 2B to more than one sentence in the legend, explaining that the method is shown schematically?

We initially believed that the BRET analysis has been explained in several of our previous papers and reviews/protocols and, therefore, felt that its detailed discussion might not be necessary beyond a cartoon and brief description. Based on these helpful comments, we have now included a more detailed description of the method and its value in our current experiments both in the Results (end of page 7) and the legend to new Fig. 1.

-- On page 8, same section, it was not clear that PI4P levels following VT01454 treatment did decrease to "below initial levels" after the initial sharp decline; the decline seems barely significant. But that class I PITPs do contribute Rab7+ compartments was clear when the cells were treated with OSBP-inhibitor, so their conclusion that class I PITP's play a role is well justified.

We have tried to explain these results better in the revised manuscript. The addition of VT01454 causes the liberation of the PI4P-binding sensors from the plasma membrane (PM) because of its robust inhibition of PM PI4P levels. This extra amount of sensor causes an increased signal in the Rab7-positive compartment. This effect is similar to what is observed when the PI4KA enzyme (the enzyme that makes PI4P in the PM) is inhibited. However, this Rab7-associated PI4P BRET signal remains high when the PI4K is inhibited (see our publications Baba et al. PMID: 31368593, Fig.1C, D or Sengupta et al. PMID: 30626625, Fig. 7B) (also attached here for the Reviewer, although in this shorter time window it may not be best judged), whereas after PITP inhibition it does not remain elevated but shows a sharp decline (we believe the decline is quite obvious in Fig. 2F, red trace). That means that the PI4P level of the Rab7 compartment decreases, even though there is plenty of unbound PI4P-binding sensor available.

--Right after, they state that there is only a minor effect on PI3P pools in Rab5+ compartments. The authors later speculate that PI levels in early endosomal compartments most likely rely heavily on plasma membrane levels--perhaps they might already provide some kind of interpretation here?

We only speculated about the possible sources of PI other than coming from PI transferred direct from the ER. We removed this speculation and only state that more experiment will be needed to sort out this question (which we continue to work on).

--This section as well as the section "Expression of drug resistant PITPs can rescue the lipid changes caused by VT01454" might benefit from summary sentences at the conclusion (in addition to the section titles).

We thank the Reviewer for these suggestions. We have now added a summary conclusion to each of these sections.

--The reviewer is very confused as to the purpose of the section "Diacylglycerol promotes membrane association of the "open" conformation of class I PITP's". Many PITP constructs are investigated, but there already seems to be a lot known regarding behavior in vitro. Is the new information that the constructs exhibit expected behavior in vivo? Perhaps the section is more intelligible to readers knowledgeable on nitty-gritty details of PITP function; again, this section seems to be written for experts, in this case PITP, rather for an audience generally interested in membrane homeostasis more generally.

It is also not clear how this section fits in with the rest of the manuscript, except characterization at the very end that indicates that VT01454 binding is consistent with stabilizing an open conformation that binds membranes. It would be helpful if the authors could explain how studies on pages 9+10 other than those pertaining to VT014154 are relevant to the story they are telling (or are they just packing in a whole bunch of experiments that they do not know how else to publish?)...

We believe that the membrane association of PITPs is an important question that has, indeed, been studied only in vitro and is still not fully understood in the literature. We trust that our new finding that DAG plays a role in this process is rather important, and also raises the question whether DAG rich membranes are where the PITPs are more likely to assume an open conformation and increase their lipid-exchange activity. We certainly do not think that these studies are just a "bunch of studies that we do not know where else to publish". We are strongly convinced that they are important in understanding how these proteins interact with membranes and what lipids possibly facilitate their lipid exchange cycle. However, we certainly appreciate the Reviewer's criticisms. Therefore, we tried to integrate this part of the study better in the revised manuscript with a clearer rationalization. As part of this effort, we have broken up the original long description of these experiments into three parts in the Result section with better articulated aims and conclusions. We also changed the order in which the results are presented, linking the structural work closely to the membrane-binding segments.

Also, throughout the whole section, the reviewer was left wondering how the C-terminus of the PITPs so much affects membrane localization. Perhaps an explanation could be offered?

We have added an explanation about how this short C-terminus can affect the function of the protein in the Discussion (top of page 21).

Dear Tamas,

Thank you submitting a revised version of your manuscript. It was sent to the same three reviewers that originally appraised your work; their comments are attached to the bottom of this email. As you will see, all three referees are satisfied with the changes you made. Before we can move forwards towards publication of your manuscript, though, there are some remaining editorial points which need to be addressed. In this regard, would you please:

- acknowledge funding from - Z01: HD000196-25 in the manuscript file, and from the National Institute Virology and Bacteriology (Programme EXCELES, Project No. LX22NPO5103) - Funded by the European Union - Next Generation EU, the Academy of Sciences of the Czech Republic RVO: 61388963 in our online submission system,
 - include up to five keywords,
 - include links to publicly available structural and mass spectrometric (if available) data sets,
 - change the title of the 'Conflict of Interests' statement to the 'Disclosure and Competing Interests Statement',
 - remove the author credit section from the manuscript,
 - replace references to 'data not shown'; these may either be with 'unpublished observations' together with the name of the person who made the observation in brackets, or by showing the data themselves in an appendix figure,
 - include a callout for Fig. 3E and replace "Extended View Table 1" with "Table EV1",
 - return to us a completed author checklist with the manuscript files (please let us know whether you would like us to send you another blank checklist),
- include the supplementary information on protein purification and purification in an appendix file, and refer to this file appropriately in the manuscript text,
- upload Source Data in one folder per figure,
 - replace the autorads in Fig EV2C with higher-resolution images,
 - consider adding a 'Data Information' subheading before 'Grand averages' in the legends for Figures 2 and 3,
 - explain panels A-D in the legend for Figure 8,
 - define sample sizes in the legend for Figure 8, and for EV Figure 1b and EV Figure 2d (here you may use a bar to indicate the range of measurements),
 - include a scale bar for the images in EV Figure 2A and 2B,
 - remove the blank page on p. 26,
 - correct 'Table I' to 'Table 1' in the appropriate legend, and
 - remove the Extended View Table and its legend from the manuscript and upload them as a separate file using the callout 'Table EV1'

We include a synopsis of the paper (see <http://emboj.embopress.org/>). Please provide me with a two-sentence general summary statement and 3-5 bullet points that capture the key findings of the paper.

We also need a summary figure for the synopsis. The size should be 550 wide by [200-400] high (pixels). You can also use something from the figures if that is easier.

I have prepared the following edits to improve the readability of the short paragraph referred to by Referee #1. Please use them at your discretion:

'Taken together, these data showed that VT01454 can promote stabilization of the open conformation of Class I PITP proteins, facilitating their interactions with the membrane. This increase in PITP-membrane interaction was not, however, as large those seen after C-terminal truncations were performed. Surprisingly, however, despite this loss of membrane interaction, C-terminally truncated PITPs still remain partially active. In contrast, the hydrophobic residues in the lipid-exchange loop appear to be dispensable, at least under our overexpression assay conditions.'

I look forward to receiving these changes. EMBO Press is an editorially independent publishing platform for the development of EMBO scientific publications.

Best wishes,

William

William Teale, PhD
Editor
The EMBO Journal
w.teale@embojournal.org

We realize that it is difficult to revise to a specific deadline. In the interest of protecting the conceptual advance provided by the work, we recommend a revision within 3 months (28th May 2024). Please discuss the revision progress ahead of this time with the editor if you require more time to complete the revisions. Use the link below to submit your revision:

Referee #1:

The authors have for the most part addressed my concerns.

The experiments on membrane association of PITPs and on the structural changes upon VT binding are now more clearly presented, but as a lot of different mutants are used, it's still a bit difficult to keep track of what are the different mutations and their effects on ligand/membrane binding/functional readouts; a summary with a structure showing all the different residues may be helpful.

I also have difficulties parsing the concluding paragraph on pg. 15, especially the last sentence:

"These data together showed that VT01454 can promote stabilization of the open conformation of the Class I PITP proteins, to facilitates membrane interactions, although it does not enhance membrane binding to the same extent as do C-terminal truncations of the PITPs. Surprisingly, however, C-terminally truncated PITPs still remain partially active, whereas the hydrophobic residues in the lipid-exchange loop appear to be dispensable at least under our overexpression assay conditions."

Referee #2:

My concerns have been addressed.

Referee #3:

The authors have addressed my concerns, particularly in adding clarifications to this very dense and informative manuscript aimed at improving readability for a more general audience. This work significantly advances our understanding PI dynamics within the cell and the manuscript is well and beautifully written. I fully support publication.

All editorial and formatting issues were resolved by the authors.

Dear Tamas,

I am pleased to inform you that your manuscript has been accepted for publication in the EMBO Journal.

Congratulations to you and your group! I will be really glad to see this insightful and very careful study in The EMBO Journal.

Best wishes,

William

William Teale, PhD
Editor
The EMBO Journal
w.teale@embojournal.org
